



# Seasonal variability in Antarctic ice shelf velocities forced by sea surface height variations

Cyrille Mosbeux[1,4], Laurie Padman[2], Emilie Klein[3], Peter D. Bromirski[1], Helen A. Fricker[1]

[1] Institute of Geophysics and Planetary Physics, Scripps Institution of Oceanography, UC San Diego, La Jolla, California, USA

[2] Earth and Space Research, Corvallis, OR, USA

[3] Laboratoire de Géologie – CNRS UMR 8538, École normale supérieure – PSL University, Paris, France

[4] Univ. Grenoble Alpes/CNRS, IGE, Grenoble, France

*Correspondence to*: Cyrille Mosbeux (cyrille.mosbeux@univ-grenoble-alpes.fr)

**Abstract.** Antarctica's ice shelves resist the flow of grounded ice towards the ocean through "buttressing" arising from their contact with ice rises, rumples, and lateral margins. Ice shelf thinning and retreat reduces buttressing, leading to increased delivery of mass to the ocean that adds to global sea level. Ice shelf response to large annual cycles in atmospheric and oceanic processes provide opportunities to examine how environmental changes affect dynamics of both ice shelves and the buttressed grounded ice. Here, we explore whether seasonal variability of sea surface height (SSH) can explain observed seasonal variability of ice velocity. We investigate this hypothesis using several time series of ice velocity from Ross Ice Shelf (RIS), satellite-based estimates of SSH seaward of the RIS front, ocean models of SSH under and near RIS, and a viscous ice sheet model. The observed annual changes in RIS velocity are of order 1-10 metres per year (roughly 1% of mean flow). The ice sheet model, forced by the observed and modelled range of SSH of about 10 cm, reproduces the observed velocity changes when visco-elastic effects near the grounding line are parameterized in our viscous model. The model response is dominated by grounding line migration, but with a significant contribution from SSH-induced tilt of the ice shelf. Improvements in measurements and models of SSH, including under ice shelves, combined with additional long-term GNSS records of ice shelf velocities, will provide further insights into longer term ice shelf and ice sheet response to future changes in sea level.

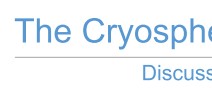


## 1 Introduction

The Antarctic Ice Sheet discharges mass via outlet glaciers and ice streams flowing into the ocean across the grounding lines, forming ice shelves several hundreds of metres thick surrounding about half of the Antarctic coastline (Allison et al., 2011; Fretwell et al., 2013). Ice shelves play critical roles in ice sheet dynamics by providing back-stresses that impede the gravity-forced flow of grounded ice towards the grounding line (Thomas, 1979). Ice shelf extent, thickness and mass can vary over time (e.g., Cook & Vaughan, 2010; Paolo et al., 2015; Adusumilli et al., 2020), leading to changes in ice velocity for both grounded and floating ice (e.g., Scambos et al., 2004; Fürst et al., 2016; Reese et al., 2018; Gudmundsson et al., 2019). Persistent ice shelf thinning or retreat over years or decades can lead to a significant increase in the rate of mass loss of grounded ice (e.g., Velicogna et al., 2014; Joughin et al., 2014; Gudmundsson et al., 2019; Smith et al., 2020), and an associated increase in the rate of Antarctica's contribution to global sea level.

Time series of ice velocity ($\mathbf{u}_{ice}$) from Global Navigation Satellite System (GNSS) receivers mounted on grounded and floating ice are, typically, of fairly short duration, limited to ~1-3 months over austral summer. These short records reveal a strong tidal-band signal (e.g., Makinson et al., 2011) but cannot resolve annual cycles. However, a few longer GNSS records, and satellite-based estimates of $\mathbf{u}_{ice}$, show variability on intra-annual (monthly to seasonal) time scales. Given that the seasonal cycle dominates variability in atmospheric and oceanic forcing of ice shelves, understanding how this forcing cycle affects ice shelf flow may provide important insights into how ice shelves and ice sheets will respond to the weaker but more persistent forcing at longer time scales, from interannual variability (e.g., Dutrieux et al., 2014; Paolo et al., 2018) to multi-decadal trends (Jenkins et al., 2018).

Two mechanisms have been proposed to explain seasonal variability of ice shelf flow, linked to seasonal variability in (i) basal melt rates and (ii) sea ice. Klein et al. (2020) investigated the hypothesis that a seasonal cycle of spatially-varying basal melt rates on Ross Ice Shelf (Tinto et al., 2019; Stewart et al., 2019) might result in seasonality of $\mathbf{u}_{ice}$; however, their modelled variability of $\mathbf{u}_{ice}$ was much smaller than GNSS measurements indicated. Greene et al. (2018)



proposed that changes in buttressing from sea ice could explain the satellite-derived seasonal cycle
of Totten Glacier's ice shelf; however, their uncertainties in satellite-derived intra-annual $\mathbf{u}_{ice}$
estimates were large, and the mechanism of ice shelf buttressing by sea ice is poorly understood.
In this paper, we investigate an alternative hypothesis: *Seasonal variability of sea surface height*
*(SSH) modifies ice velocity through a combination of sea surface tilt and changing basal stresses*
*at the grounding zone.* This hypothesis is motivated by an extension of the role of tides on ice
shelves and grounded-ice motion (Gudmundsson et al., 2007; Gudmundsson et al., 2013; Brunt
and MacAyeal, 2014; Rosier et al., 2020), evidence from open ocean satellite altimetry that SSH
around Antarctica has a pronounced seasonal cycle (Armitage et al., 2018; Rye et al., 2014). and
the recent development of ocean models from which estimates of seasonal variability of SSH under
ice shelves can be extracted. We explore our hypothesis by running a viscous model of the ice
sheet and ice shelf in the Ross Sea sector with forcing from the modelled seasonal cycle of SSH
under Ross Ice Shelf, and comparing the model output with GNSS time series of ice shelf velocity.
We selected Ross Ice Shelf because variability in ice shelf mass balance at longer time scales is
known to be small (Das et al., 2020; Adusumilli et al., 2020. In addition, there are several GNSS
records exceeding one year in length that reveal intra-annual variability. We show that the ice sheet
model reproduces the observed intra-annual variability of the GNSS records if visco-elastic effects
near the grounding line are parameterized in our viscous model. This response is dominated by
grounding line migration, but with a significant contribution from SSH-induced tilt of the ice shelf.
## 2 Data and Models
We explore our hypothesis using a combination of in situ and satellite-derived observations, and
ocean and ice sheet modelling. We take advantage of several existing GNSS records from Ross
Ice Shelf (RIS) collected during various field campaigns (Sec. 2.1), focusing on the ones that are
sufficiently long to identify intra-annual velocity variations. We combine these records with
estimates of intra-annual variations in SSH fields for the open ocean in front of the ice shelves
from an existing satellite altimetry data set and from ocean models that include ice shelves (Sec.

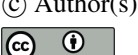



2.2). We then compare the GNSS records to an ice flow model forced with the varying SSH from
the ocean models (Sec. 2.3).

### 2.1    GNSS Data

We use several long (roughly one year or longer) time series of ice shelf motion from GNSS
deployments on RIS (**Fig. 1**). These records were collected during different time intervals between
2014 and 2019 (**Table 1**). GNSS data from all stations were processed with a Precise Point
Positioning (PPP) approach (Zumberge et al., 1997; Geng et al., 2012).
**DRRIS 2015-2016:** An array of 13 GNSS stations was deployed on RIS from November 2015 to
December 2016 as part of the Dynamic Response of the Ross Ice Shelf to Wave-Induced
Vibrations (DRRIS) project (Bromirski and Gerstoft, 2017; Klein et al., 2020). Three stations were
deployed along the ice front and nine along a flowline from the central ice front station to about
400 km upstream. One station (RS03) was located 100 km to the west of the along-flowline array
and another (RS08) was on grounded ice on the western margin of Roosevelt Island. Only one
station (DR10) recorded position data for a full year; however, the intra-annual signals in positions
and velocities at the other DRRIS stations on floating ice were highly correlated with DR10
observations (Klein et al., 2020, their Figure 6).
**WISSARD 2014-2016:** An array of GNSS stations was deployed as part of the Whillans Ice
Stream Subglacial Access Research Drilling (WISSARD; Siegfried et al., 2014; Tulaczyk et al.,
2014) project. We used the record from station "GZ19" located about 3 km offshore of the
Whillans Ice Stream grounding line, that acquired data between November 2014 and November
2016 (Begeman et al., 2020).
**Antarctica PI Continuous network 2017-2019:** Two GNSS stations (BATG and LORG)
acquired data in the northwestern RIS. We obtained the time series for these sites from the GNSS
database processed by the Nevada Geodetic Laboratory (NGL; Blewitt et al., 2018). Station BATG
was located about 100 km east of Minna Bluff and acquired data from February 2017 to August
2018. Station LORG is located about 100 km east of Ross Island and about 90 km from BATG;





the station recorded from November 2018 to November 2019 with a few interruptions, for a total
of 289 days. The vertical components of tidal variability at these stations were reported by Ray et
al. (2020).

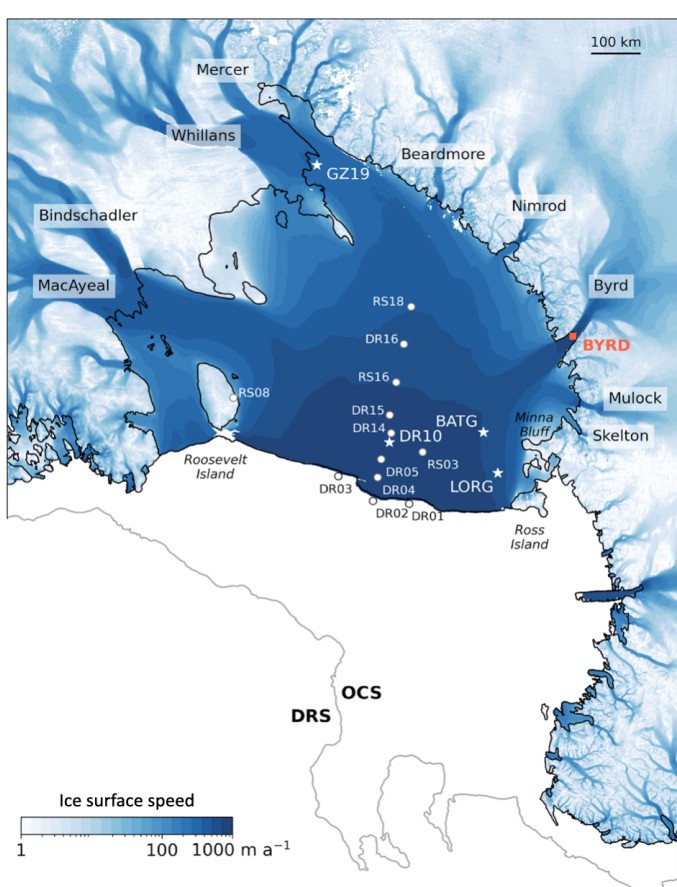


**Figure 1.** Map of Ross Ice Shelf and its surrounding principal outlet glaciers and ice streams. The locations
of GNSS stations used in this study and their names are indicated; see **Table 1** for more details. Our focus
is on long time series from DR10, BATG, LORG and GZ19 (white stars). BYRD (orange square) is not a
GNSS site but identifies the area analysed in **Fig. 9e**. The background image shows time-averaged surface
velocities measured by satellites (Rignot et al., 2016). The grounding line and the ice front, from Depoorter
et al. (2013), are plotted with black lines. The 1500 m isobath, separating regions defined as the open
continental shelf (OCS) and the deep Ross Sea (DRS), is plotted in dark grey.



**Table 1.** Station latitudes and longitudes at time of deployment, mean speed, database/project which
collected the data, duration (number of days of available data), and periods of deployment for GNSS
stations. The primary stations used in this study are indicated in bold.

| GNSS Station | Longitude | Latitude | Mean speed (m/a) | Project/database | Duration (days) | Period |
|---|---|---|---|---|---|---|
| **DR01** | -178.35 | -77.77 | 1023 | DRRIS | 197 | Nov 2015 – Nov 2016 |
| **DR02** | -178.42 | -77.82 | 1089 | DRRIS | 221 | Nov 2015 – Dec 2016 |
| **DR03** | -175.12 | -78.26 | 993 | DRRIS | 219 | Nov 2015 – Dec 2016 |
| **DR04** | -178.79 | -78.28 | 1030 | DRRIS | 214 | Nov 2015 – Dec 2016 |
| **DR05** | -179.88 | -78.63 | 987 | DRRIS | 216 | Nov 2015 – Dec 2016 |
| **DR10** | **-179.88** | **-78.96** | **937** | **DRRIS** | **331** | **Nov 2015 – Nov 2016** |
| **DR14** | 179.95 | -79.14 | 903 | DRRIS | 223 | Nov 2015 – Dec 2016 |
| **DR15** | -179.92 | -79.49 | 858 | DRRIS | 180 | Nov 2015 – Nov 2016 |
| **DR16** | -178.43 | -80.87 | 572 | DRRIS | 152 | Nov 2015 – Sept 2016 |
| **RS03** | 176.88 | -78.76 | 894 | DRRIS | 177 | Nov 2015 – Nov 2016 |
| **RS08** | -163.54 | -79.39 | 7 | DRRIS | 148 | Nov 2015 – Oct 2016 |
| **RS16** | 179.37 | -80.13 | 682 | DRRIS | 142 | Nov 2015 – Nov 2016 |
| **RS18** | 177.33 | -81.59 | 493 | DRRIS | 119 | Nov 2015 – Mar 2016 |
| **GZ19** | **-163.64** | **-84.33** | **307** | **WISSARD** | **579** | **Nov 2014 – Nov 2016** |
| **BATG** | **170.72** | **-77.57** | **670** | **NGL** | **565** | **Jan 2017– Aug 2018** |
| **LORG** | **170.03** | **-78.18** | **618** | **NGL** | **289** | **Nov 2018 – Nov 2019** |


**2.2   SSH measurements and model estimates**
SSH can be estimated using satellite radar altimetry, and monthly SSH estimates are available for
the period 2011-2016 for regions north of the Antarctic coastline and ice shelves using
measurements from the European Space Agency's CryoSat-2 radar altimeter (Armitage et al.,
2018). These SSH estimates cover fully open water (free of ice shelves) and leads in the ice pack,





but do not extend under the ice shelves. Measuring SSH variations in the ocean cavities under ice
shelves is challenging because they are small compared with other contributors to height changes,
such as uncertainties in seasonal cycles of basal mass balance (e.g., Stewart et al., 2019; Tinto et
al., 2019), snow and firn density changes (e.g., Zwally and Jun, 2002; Arthern and Wingham,
1998), and penetration of radar signals into the surface snow and firn layers (Ridley and Partington,
1988; Davis and Moore, 1993). Therefore, it is not currently possible to accurately estimate SSH
variability under ice shelves.
Instead, we investigated the representation of intra-annual variability of SSH from five existing
ocean models with thermodynamically active ice shelves (Mathiot et al., 2017; Tinto et al., 2019;
Naughten et al., 2018; Dinniman et al., 2020; Richter et al., 2020), using their SSH output relative
to the Armitage et al. (2018) open-water data set to determine the most realistic model for analyses
and to assess the likely variability of SSH under ice shelves. More information on these models,
and assessment of their performance, is provided in Supplementary Information (SI). From these
analyses we determined that the Ross Sea regional model described by Tinto et al. (2019) provides
a seasonal cycle that is most consistent with the Armitage et al. (2018) satellite-based results for
the Ross Sea continental shelf north of RIS, suggesting that it is also the best model for SSH
variability under RIS.
**2.3 Ice sheet / ice shelf model**
**2.3.1 Model summary, and initialisation**
We used the open-source ice sheet and ice flow model Elmer/Ice (Gagliardini et al., 2013), the
glaciological extension of the Elmer finite element software developed at the Center for Science
in Finland (CSC-IT). The modelling framework is similar to that described by Klein et al. (2020).
We added variability of SSH in both time and space, relative to the initial static sea level, focusing
on SSH output from the Tinto et al. (2019) ocean model as justified in SI. Our ice model uses the
vertically-integrated Shallow-Shelf Approximation (SSA; MacAyeal, 1989), a simplification of
the Stokes equations (usually used for resolving viscous flow problems) in which the ice velocity
is considered constant throughout the ice thickness. This approximation is well suited to ice shelves





and ice streams where vertical shear stresses are negligible relative to other stresses acting on the
ice. In addition, we used a linear Weertman friction law (Weertman, 1973) and a non-linear
constitutive relationship between strain rates and deviatoric stresses, classically used in ice flow
modelling and known as Glen's flow law (Glen, 1958).
Following the same procedure used by Klein et al. (2020), we initialised our model by inferring
the basal shear stress (on grounded ice) and the ice viscosity, using an inverse model that optimises
the two parameters by minimising the difference between model and observed surface ice
velocities as well as the difference between ice flux divergence and observed mass balance.
There are two main effects of SSH variability on the ice shelf velocities: (i) changes in driving
stress and (ii) changes in basal stress through grounding line migration.

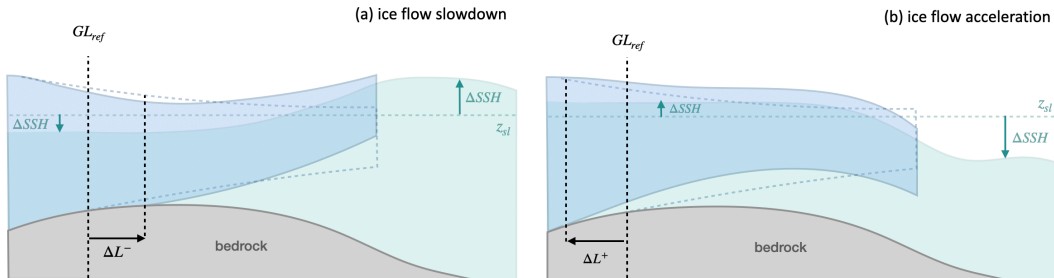


**Figure 2.** Conceptual model of the SSH effect on the ice shelf slope and grounding line position:
combination of (a) a positive ice shelf tilt and a negative $\Delta SSH$ close to the grounding line, (b) a negative
ice shelf tilt and a positive $\Delta SSH$ close to the grounding line. The average annual state of the ice shelf is
shown by dashed lines while the perturbed state is shown by plain lines.
**(i) Driving stress change**
Changes in gradients of SSH locally impact the driving stress, $\sigma_g$ (in MPa), acting on the ice flow.
This stress is a direct function of the surface gradient, $\nabla z_s$, with $z_s$ being the ice shelf surface
height (assuming solid ice from surface to base) relative to the background unperturbed sea
surface, following, e.g., Morland (1987), MacAyeal (1989) and Gudmundsson (2013):



$\sigma_g = \rho_{ice} g \, h \, \nabla(z_s + \Delta SSH).$                                                (1)
In Eq. (1), $\rho_{ice}$ is the density of ice (917 kg m$^{-3}$, assumed constant over the ice thickness), $g$ is the
gravitational acceleration (9.81 m s$^{-2}$), $h(x, y, t)$ (m) is the ice shelf thickness, and $\Delta SSH(x,y,t)$ is
the SSH perturbation. A decrease of the ice shelf seaward gradient leads to an increase in driving
stress and an ice flow slowdown (**Fig. 2a**). An increase in the ice shelf seaward gradient leads to
an increase of driving stress and an acceleration of the ice flow (**Fig. 2b**).
**(ii) Change in basal stress through grounding line migration**
SSH variations lead to changes in bed stresses in the grounding zone as they raise and lower the
ice shelf. A negative $\Delta SSH$ at the grounding line causes a downstream migration of the grounding
line, increasing the grounded-ice area and potentially slowing down ice movement through an
increase in basal drag (**Fig. 2a**). Conversely, a positive $\Delta SSH$ at the grounding line leads to an
upstream migration of the grounding line, decreasing the area affected by basal stresses and
accelerating the ice flow (**Fig. 2b**). The grounding-line migration distance ($\Delta L$) upstream and
downstream is influenced by visco-elastic deformation of the ice shelf. The mechanism has been
studied in the context of tidal deformation by treating it as an elastic and hydrostatic beam problem
(e.g., Sayag and Worster, 2011 and 2013; Walker et al., 2013). This analytical solution agrees
reasonably well with grounding line migration calculated by solving the contact problem in a
visco-elastic, tide-forced model (Rosier et al., 2014).
In a purely hydrostatic framework, the grounding line migration ($\Delta L$) depends on both the surface
and bed slopes (Eqs. B1 and B2; Appendix B) (Tsai and Gudmundsson, 2015): as surface and bed
slopes decrease, $\Delta L$ increases. This inverse relationship directly affects the magnitude of the
change in friction in the grounding zone, and also the ice flow response. The implications of
uncertainties in our knowledge of bed slope and $\Delta L$, and the mechanical processes involved, are
discussed further in Sec. 4.2.
Grounding line migration $\Delta L$ has also been treated as an elastic fracture problem, accounting for
water pressure variations at the ice base as the grounding line migrates. Using this framework, Tsai



and Gudmundsson (2015) showed that the magnitude of upstream $\Delta L$ is larger than in the
hydrostatic or purely elastic case, and depends non-linearly on parameters such as the ice thickness
and $\Delta SSH$. For thick ice (e.g., in the grounding zone of Byrd Glacier) $\Delta L$ can be more than twice
the value obtained using the hydrostatic framework, and for small $\Delta SSH$ (typically, a few
centimetres), $\Delta L$ can be as much as one order of magnitude higher than in the hydrostatic
framework.
**2.3.2 Model runs**
We ran 100 inversions of both the basal friction and the ice viscosity, constraining the fit to
observations and the degree of smoothness of the solution. From this ensemble of initial states, we
selected an optimal (in terms of velocity and ice flux divergence fit) sub-ensemble of 15 members
($\Omega_{15}$). The details of the initialisation procedure and the selection of $\Omega_{15}$ are discussed in
Appendix A1.
Using the sub-ensemble $\Omega_{15}$ as a reference, we applied monthly averaged SSH anomalies ($\Delta SSH$)
from five different ocean models (SI) as a steady-state perturbation, raising or lowering the ice
surface, and ice base and computing the flow change with respect to the reference (see Appendix
A2). For each run, we kept the ice shelf thickness $h$(x,y,t) constant and assumed that the ice shelf
and the grounding line location adjusts instantaneously to the $\Delta SSH$.
To assess the importance of methods for representing grounding line migration in our viscous ice
sheet model, we ran three different parameterisations of $\Delta L$  (for a total of 225 simulations = 15
members x 5 SSH models x 3 grounding line parametrizations), as follows:
(i)      $\Delta L_{B2}$: based on the hydrostatic equilibrium of the grounding line and Bedmap2 (a gridded

products describing surface elevation, ice-thickness and the basal topography of the

Antarctic; Fretwell et al., 2013) bed slopes at the grounding line.

(ii)     $\Delta L_C$ (constant bed slope): a significantly larger migration that corresponds to values used

by Rosier and Gudmundsson (2020) for their study of Filchner-Ronne Ice Shelf when



treating the grounding line migration with elastic fracture mechanics introduced by Tsai
and Gudmundsson (2015).
(iii)    $\Delta L_{B2L}$: also a larger value of grounding line migration but accounting for the Bedmap2
surface and bed slope variations along the grounding line.
To account for subgrid-scale migration of the grounding line, our model implementations
parameterise $\Delta L$ as a change in friction, rather than as a change in floatation state at specific grid
nodes (Appendix B).

## 3    Results and Discussion

We first review the intra-annual variability of ice flow recorded by the GNSS receivers on RIS
(Sec. 3.1.1) and the measured (Armitage et al., 2018) and modelled (Tinto et al., 2019) seasonal
cycles of SSH for the Ross Sea including under RIS (Sec. 3.1.2). We then compare the variability
of driving stresses due to SSH anomalies and grounding line migration (Sec. 3.2), and the effect
of both processes on the ice speed flow (Sec. 3.3).

### 3.1    Intra-annual signals in GNSS displacement and SSH records
### 3.1.1    GNSS displacement

All long-duration GNSS stations on RIS (Sec. 2.1) show variability in horizontal displacement on
various time scales including diurnal (~1-day period), fortnightly (~2-week period) and intra-
annual (**Fig. 3**). As reported by Klein et al. (2020), data from the DRRIS stations show evidence
of an annual cycle with a displacement anomaly amplitude of about 1 m, alternating between a
negative trend during December-May and a positive trend during June-November. GZ19 shows
no apparent annual cycle, but its displacement shows a similar range of variability during the 2-
year record. The time series at BATG, which is not concurrent with the DRRIS stations and GZ19,
shows a smaller amplitude range (about 0.2-0.3 m) that appears to have a periodicity of about 6
months. The LORG time series in 2019 shows a similar pattern to BATG in 2018.

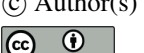
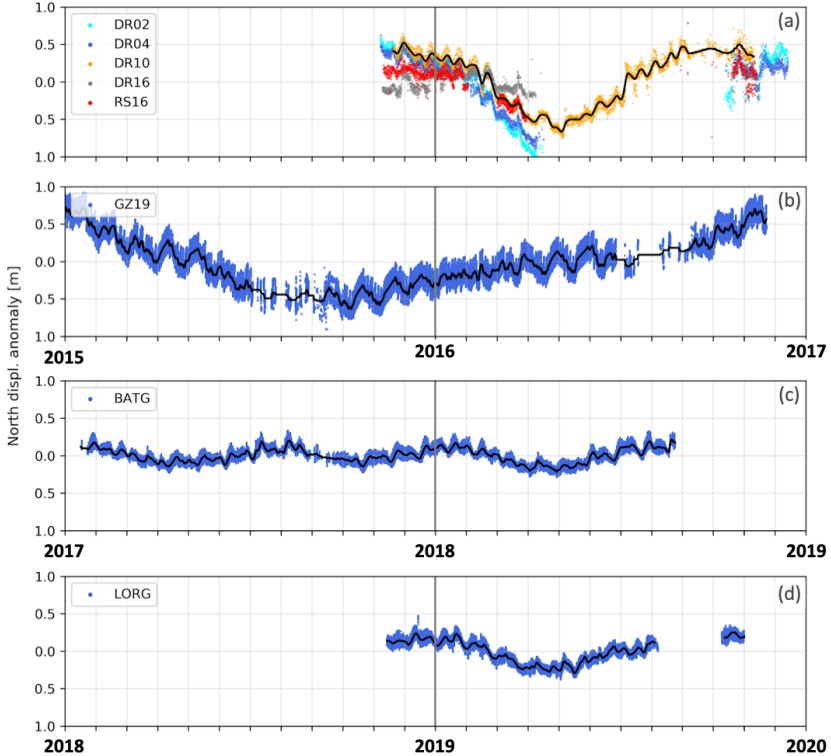

**Figure 3.** GNSS horizontal displacement anomalies in the north direction (approximately parallel to the time-averaged flow) for GNSS stations used in this study. Time interval for each panel is two years; however, years differ between panels. (a) DR02, DR04, DR10, DR16 and RS16 (for legibility, other DRRIS sites are not shown here but exhibit a similar trend; the complete array can be found in Klein et al. (2020)); (b) GZ19; (c) BATG and (d) LORG. Note that (a) and (b) are plotted on the same time scale while (c) and (d) have 2-year and 3-year shifts with respect to the two upper panels. The black lines are smooth versions of displacement anomalies with a 1-day Gaussian RMS width.

The diurnal lateral displacement signal is caused by the fundamental tides of the region, which are almost entirely diurnal (e.g., Padman et al., 2003; Ray et al., 2020). We attribute the fortnightly signal in displacement at all GNSS sites (and, possibly, also the ~6-month periodicity at BATG and LORG) to nonlinear response of the ice sheet and ice shelf to variability of the tidal range,


leading to visco-elastic flexural adjustments of the ice sheet at the grounding zone as the range of
the diurnal tide varies through the fortnightly spring-neap modulation (e.g., Rosier et al., 2020).
We removed the fortnightly tide-forced variability by filtering to monthly and longer time scales;
however, any ~6-month tidal signal remains as a source of noise in our interpretation of intra-
annual ice shelf flow changes driven by non-tidal SSH variability.
**3.1.2 Satellite-derived and modelled SSH**

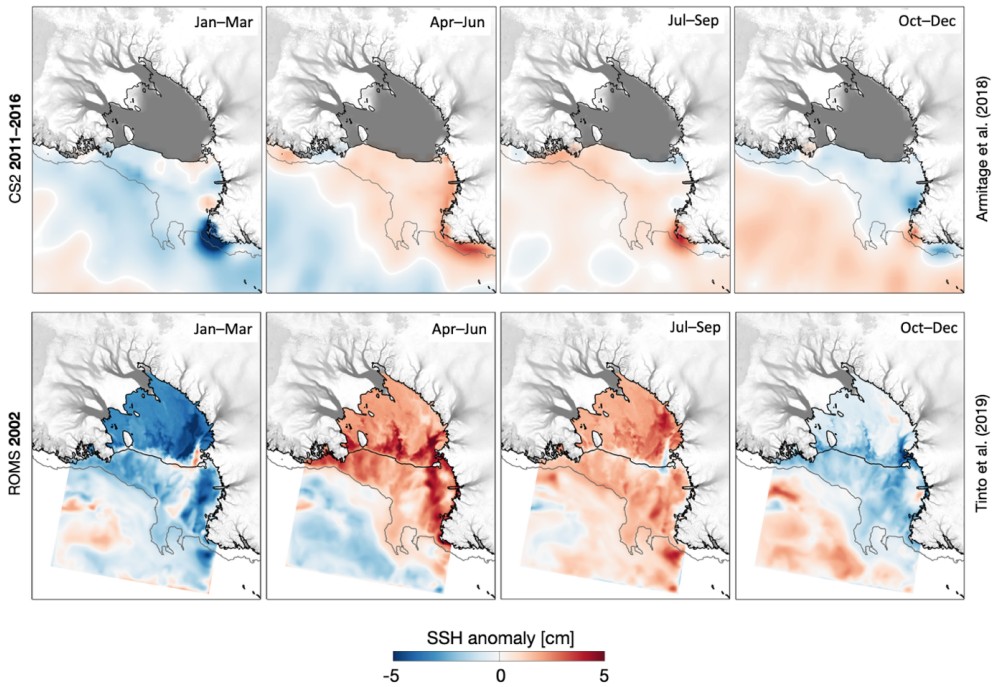


**Figure 4.** Seasonal sea surface height deviation from the annual mean (ΔSSH): (top row) satellite
observations averaged over the period 2011–2016 ($\Delta SSH_{CS2}$, Armitage et al., 2018) and (bottom row)
modelled for the period 2002 ($SSH_{2002}$, Tinto et al., 2019). The ice front and grounding line are represented
by black lines. The outer edge of the open continental shelf (OCS) is along the 1500 m isobath, shown with
a grey line. Ice speeds are shown in shades of grey, with darker shades being faster.



The seasonal cycle of $\Delta SSH$ in satellite-derived SSH fields around Antarctica, for the period 2011-
2016, shows a typical range of about 5 cm on the Open Continental Shelf (OCS; see **Fig. 1**) of the
Ross Sea, and comparable changes offshore in the Deep Ross Sea (DRS); see **Fig. 4**, top row, and
**Fig. 5**). For the OCS, a clear positive SSH anomalies occur in winter (April-September). The Tinto
et al. (2019) model, based on annually repeating forcing for 2002, shows similar phasing of the
$\Delta SSH$ cycle (**Fig. 4, bottom row; Fig. 5a**) but with larger amplitude than for the satellite-derived
fields. The qualitative agreement between the model and the observations offshore of RIS provides
support for the use of this ocean model for predicting SSH variability under RIS, even though the
ocean model does not overlap in time with either the observed SSH fields or the GNSS
observations.

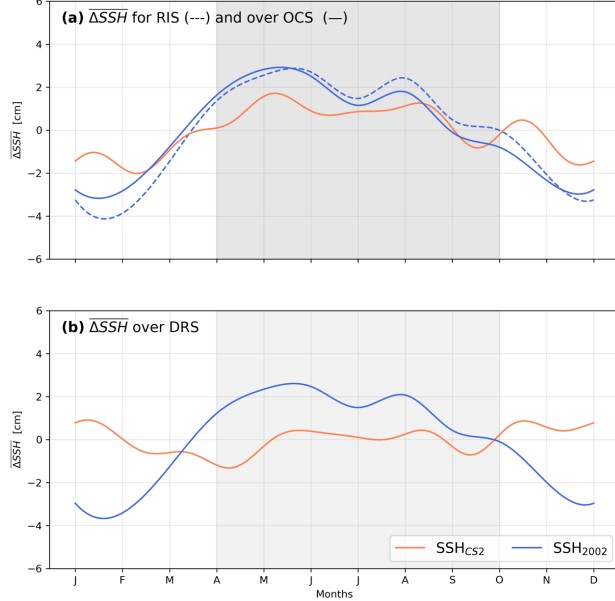


**Figure 5.** (a) Annual cycle of monthly mean $\Delta SSH$ over the open continental shelf (OCS – plain lines) and
beneath the ice shelf (RIS – dotted lines) for $SSH_{2002}$ (blue), and for CryoSat-2 measurements ($SSH_{CS2}$,
red) averaged over 2011-2016, for the open continental shelf (OCS) only. (b) Mean $\Delta SSH$ for the deep Ross
Sea. The grey shade shows the winter period. See **Fig. S3** for similar comparisons that include all available
ocean models of SSH.



## 3.2 Comparing driving stress change and grounding line migration

RIS thickness decreases from ~800 m close to the grounding line to ~300–400 m at the ice front, over a distance of ~800 km (see Tinto et al. (2019), their Fig. S2a). This results in mean thickness and surface gradients of about $5 \times 10^{-4}$ and $5 \times 10^{-5}$ respectively. Since we are interested primarily in along-flow variations of ice velocity, we calculate the along-flow Lie derivatives (Yano, 2020) of the ice shelf surface height ($\nabla_{\hat{u}} z_s$) and $\Delta SSH$ ($\nabla_{\hat{u}} SSH$). Values of $\nabla_{\hat{u}} z_s$ range from $10^{-5}$ to $10^{-2}$ over most of the ice shelf **(Figs. 6a and S4a)**. Gradients of $\Delta SSH$ in Tinto et al. (2019; $SSH_{2002}$) can reach $10^{-6}$ to $10^{-5}$ in February **(Figs. 6b and S4b)**. This means that local tilting of the ice shelf by $\nabla_{\hat{u}} SSH$ can modify the local driving stress of the ice shelf (Eq. (1)) typically by 0.1-1%, and sometimes up to several percent, with substantial spatial variability **(Fig. 6c)**. $\nabla_{\hat{u}} SSH$ also varies by month (not shown). For example, in February, about 30% and 6% of the ice shelf experiences a fractional change of driving stress exceeding 0.1% and 1%, respectively **(Fig. 6d)**. The largest fractional change in driving stress occurs away from the grounding line where the ice surface height gradients are smaller than closer to the grounding line.

The complex spatial variability of the along-flow derivatives of $\Delta SSH$ **(Fig. 6b)** arises from changes in orientation and magnitude of the sub-ice-shelf circulation relative to ice flow. This circulation is itself complex: see, e.g., Supplementary Video 1 in Tinto et al. (2019).

For most months there is a strong along-flow gradient in SSH close to the ice front **(Fig. 4b and Fig. 6b)**, which directly impacts driving stress (Eq. (1)). These variations in driving stress lead to ice velocity changes, which we present as anomalies with respect to the annual average velocity field. In general, months with a regionally-averaged negative $\Delta SSH$ (e.g., January–March period in **Figs. 4 and S2**) that slows ice flow as the grounding line migrates seaward also experience a relative uplift of the surface close to the ice front, leading to an additional slowdown **(Fig. 2a)**. Conversely, the months experiencing a regionally-averaged positive $\Delta SSH$ generally show a relative surface drop close to the ice front and an upstream migration of the grounding line, both contributing to an acceleration of the ice shelf **(Fig. 2b)**.


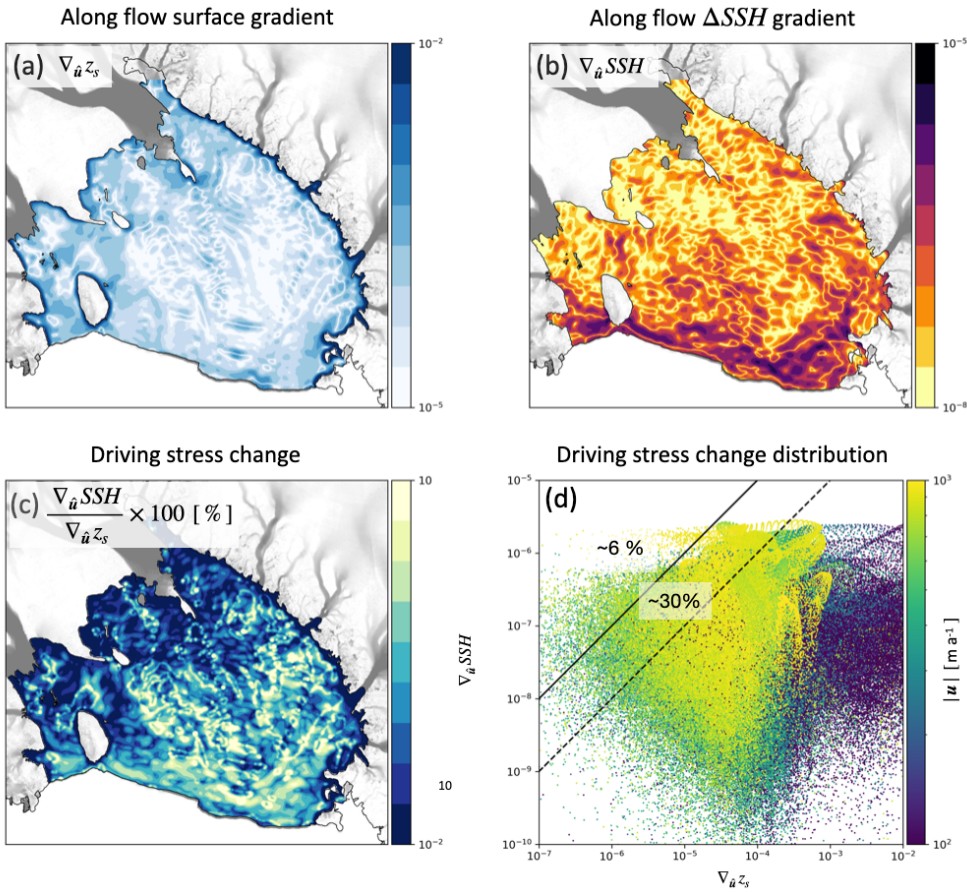

**Figure 6.** Comparison of ice shelf surface gradients and SSH gradients from $SSH_{2002}$ (Tinto et al., 2019),
both calculated in the direction of ice flow (û) in February. (a) Ice shelf surface gradient ($\nabla_{\hat{u}} z_s$), (b) SSH
gradient ($\nabla_{\hat{u}} SSH$), and (c) their ratio. Gradients are filtered with a 5-km standard-deviation Gaussian
smoothing. (d) Gradient values, for each $1 \times 1$ km cell, plotted as a function of each other. The colormap
represents the ice flow speed. 6% and 30% of the model nodes over the ice shelf experience a driving stress
variation of more than 1% (left of the plain line) and 0.1% (left of the dashed line), respectively.


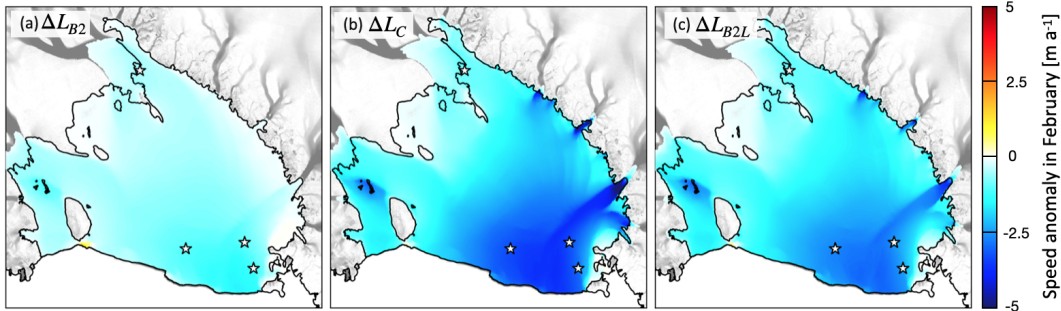

**Figure 7**. February anomaly in velocity ΔU, averaged over an ensemble of 15 initial states ($\Omega_{15}$), formed as the difference between the annual average for $\Delta SSH_{2002}$ and the three different parameterisations of the grounding line migration: (a) $\Delta L_{B2}$, (b) $\Delta L_C$ and (c) $\Delta L_{B2L}$ (see Sec. 2.3.2). The locations of DR10, GZ19, BATG and LORG (identified in Fig. 1) are indicated by white stars. The grounding line and the ice front are shown by black lines. The background annual average flow velocity for grounded ice is plotted in shaded grey, with darker grey being faster.

The modelling indicates that the amplitude of the grounding line migration, $\Delta L$, is the primary control on the amplitude of the seasonal velocity signal. In February, for example, the model ensemble using $\Delta L_{B2}$ predicts the smallest amplitude of velocity deviation of the three cases, with $\Delta U_{B2} \sim -1$ m a$^{-1}$ over most of the ice shelf (**Fig. 7a**). Larger values of $\Delta L$ (parameterisations $\Delta L_C$ and $\Delta L_{B2L}$) allow the grounding line to move farther downstream during summer, leading to deviations $\Delta U \sim -3$ m a$^{-1}$ in the centre of the ice shelf (**Fig. 7b,c**). The largest differences between the effects of $\Delta L_C$ and $\Delta L_{B2L}$ are generally found close to the grounding line in the deep and narrow fjords such as the floating extension of Byrd Glacier where $\Delta L_C$ leads to a slowdown $\Delta U_C < 5$ m a$^{-1}$ compared with $\Delta U_{B2L} \sim 3$ m a$^{-1}$ (**Fig. 7b,c**). These are regions where true bed slopes are steeper than the average around the RIS perimeter, and which are also more sensitive to the initial state as the ensembles show a larger standard deviation in these areas with respect to the rest of the domain (**Fig. 8**, bottom row).

We regard the $\Delta L_{B2}$ parameterisation, which yields small grounding line migration, as an approximation of ice shelf response to SSH gradients alone.





### 3.3 Seasonal cycle in ice flow


All ensembles forced with $\Delta SSH_{2002}$ exhibit a maximal seasonal negative flow speed anomaly
during summer and maximal positive anomaly during winter (**Fig. 8**); however, $\Delta L_{B2}$ simulations
tend to switch to positive anomalies later than simulations using $\Delta L_C$ and $\Delta L_{B2L}$. Simulations
using $\Delta L_{B2}$ produce maximal amplitudes of speed anomaly at the ice front that progressively
decrease farther upstream while $\Delta L_C$ and $\Delta L_{B2L}$ produce maximal speed anomaly amplitudes in
the deep fjords along the base of the Transantarctic Mountains. The amplitudes of speed anomalies
of $\Delta L_{B2}$ are about 2–4 times smaller than for $\Delta L_C$ and $\Delta L_{B2L}$ simulations, depending on location.
To validate the results of the three grounding zone parameterisations, we extracted the modelled
ice velocity anomalies at the GNSS locations and compared these to velocity variations (**Fig. 9**)
estimated from the time derivative of measured displacement anomalies (**Fig. 3**).
At DR10, the range of the observed velocity anomaly ($\Delta U$) was about 10 m a$^{-1}$ with a minimum in
February-March and a maximum in July (**Fig. 9a**). The other DRRIS GNSS stations located in the
centre of the ice shelf did not record during austral winter (see **Fig. 5a**), preventing us from
properly identifying the timing of maximum velocity for these stations. The $\Delta L_C$ and $\Delta L_{B2L}$
ensembles both give similar $\Delta U$ estimates that qualitatively similar to observations, with velocity
variations about 50 to 70% of the observed amplitude and minima and maxima in summer and
winter, respectively. The $\Delta L_{B2}$ grounding-zone parameterisation has a much lower amplitude and
gives a maximum velocity in October, about 2 months later than the other ensembles and 4 months
later than the observations. However, the timing of the summer $\Delta U$ minimum is close to the
observations and the other grounding-zone parameterisations. Expanding our analysis to the entire
GNSS array of DRRIS, similar seasonal phasing occurred at each GNSS station located
approximately along the central flowline of the ice shelf. $\Delta U$ amplitude generally decreases with
increasing distance from the ice front (**Fig. 10**), although with some variability that may result
from proximity of the DRRIS array to the Byrd Glacier flow and its impact on RIS flow.
At GZ19, close to the grounding line of Whillans Ice Stream, there is no seasonal cycle visible in
the GNSS observations of displacement anomaly (**Fig. 3b**). The measured velocity anomaly (**Fig.**





**9b**) shows an overall slowdown, consistent with previous observations of slowdowns of Whillans
and Mercer ice streams and the adjacent region of RIS over the last decades (e.g., Joughin et al.,
2005; Thomas et al., 2013), and shorter periods of deceleration and acceleration that could be due
to the inherent variability of the two ice streams (e.g., Winberry et al., 2009). This trend was not
captured by our ice flow models, which do not account for varying forcing other than the annual
cycle of SSH. The modelled anomalies at GZ19 are weak, with $\Delta U_{B2} \sim \pm 0.5$ m a$^{-1}$ and $\Delta U_C \sim$
$\Delta U_{B2L} \sim \pm 1$ m a$^{-1}$ over the year.
At station BATG, about 100 km east of Minna Bluff, the velocity time series shows an
approximately six-month periodicity, with a $\Delta U$ range of about 2.5 to 3 m a$^{-1}$ (**Fig. 9c**). $\Delta L_{B2}$
provides a poor fit to these observations, in both $\Delta U$ amplitude and phase, with the amplitude better
reproduced by $\Delta L_C$ and $\Delta L_{B2L}$. However, the pattern of observed velocity anomaly changes
between the first and second year of the record. In the first year, the six-month cycle shows a large
velocity drop in July-August (reaching a minimum in September), corresponding to the second
minimum of the year. In the second year, the observed velocity reached a maximum in May and
remained relatively high until the end of August, fitting the modelled velocities. While the record
terminated at the end of August, this marked a particularly long plateau of high velocities (from
May to August), suggesting that the record includes a seasonal signal that is added to the six-month
cycle that we tentatively attribute to semiannual changes in tidal range (see Sec. 1).


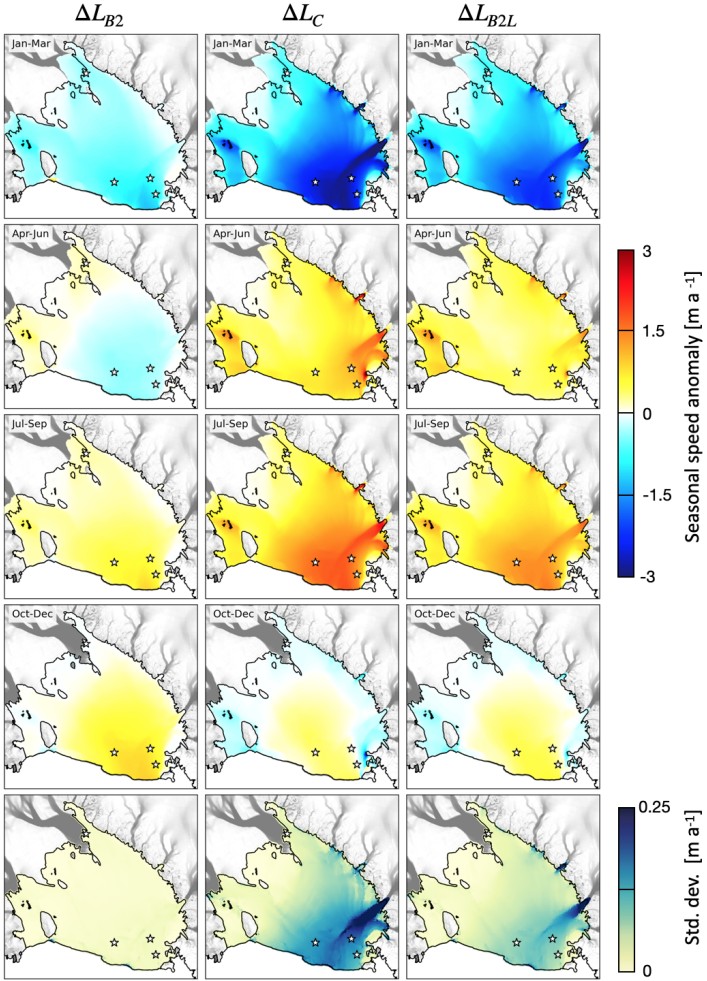


**Figure 8.** Ensemble mean seasonal (three-month average) ice flow anomaly for $\Delta SSH_{2002}$ and three parameterisations of the grounding line migration: (first column) Bedmap2 ($\Delta L_{B2}$), (second column) a constant bed slope ($\Delta L_C$), and (third column) a flatter version of Bedmap2 ($\Delta L_{B2L}$). The seasonal anomalies are computed from monthly model outputs. The standard deviation over each 15-member ensemble (bottom row) shows variability in space and time over the year. The locations of DR10, GZ19, BATG and LORG are indicated by white stars (identified in Fig. 1). The ice front and the grounding line are indicated by the black line. Ice surface velocities over the grounded ice are plotted with a grey scale, from white (slow flow) to dark grey (fast flow).

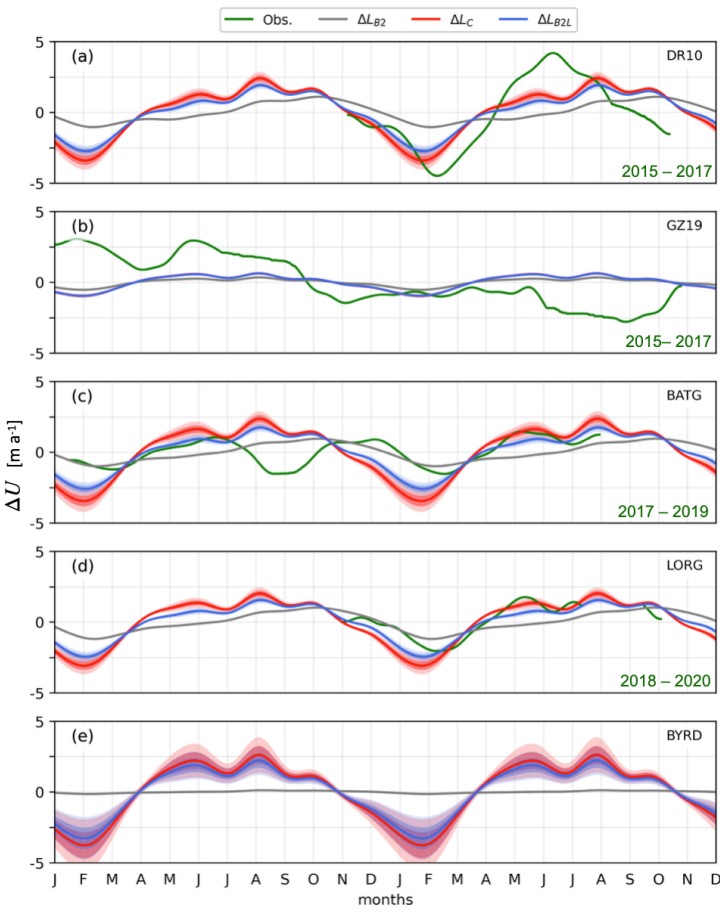

**Figure 9.** Comparison between GNSS and model velocity anomaly when applying $\Delta SSH$ values from $SSH_{2002}$ for (a) DR10, (b) GZ19, (c) BATG, and (d) LORG, and (e) at Byrd Glacier outlet (see locations on **Fig. 1**). The annual model cycle is repeated over 2 years. The average model velocity anomalies (over $\Omega_{15}$ ensembles) —$\Delta U_{B2}$ (grey), $\Delta U_C$ (red), and $\Delta U_{B2L}$ (blue) — are displayed with one and two standard deviations of the 15 estimates in each ensemble (dark and light shades, respectively). If not visible, the standard deviation is statistically insignificant. In (b), $\Delta U_C$ (red) and $\Delta U_{B2L}$ (blue) are so similar that we don't distinguish them. The observed velocities (green) are obtained as the time derivative of the measured displacement anomaly (the period of observation is given in green on each panel) from GNSS, with a Gaussian filter with a two-week standard deviation. See **Fig. S6** for similar comparisons that include all available ocean models of SSH.



The time series of $\Delta U$ at LORG (**Fig. 9d**) for the period November 2018 to November 2019 is
highly correlated (p=0.95) with the time series at BATG over the second year (from November
2017 to October 2018). The predicted velocity anomalies for $\Delta L_C$ and $\Delta L_{B2L}$ at these two stations
agree especially well with the observations over the entire LORG times series and the second year
of the BATG time series. More specifically, the model is able to reproduce the month-to-month
accelerations and decelerations and the overall longer span of positive anomalies visible in LORG
observations.

To examine the relative effect of the variations in driving stress, and basal friction through
grounding line migration, we consider a key region of RIS, the floating extension of Byrd Glacier
near its grounding line. Byrd Glacier is the fastest and the deepest outlet glacier feeding RIS, and
is the region of RIS where the outputs from the three ensembles ($\Delta L_{B2}$, $\Delta L_C$, and $\Delta L_{B2L}$) deviate
the most. Observations show that, over a time span of a few years, flow upstream of the grounding
line can increase by 10%, coinciding with the discharge of subglacial lakes lubricating the bed
(Stearns et al., 2008). At seasonal time scales, variations of Byrd Glacier remain poorly constrained
due to the lack of year-round GNSS measurements; however, Greene et al. (2020) used feature
tracking in satellite imagery to estimate ice velocities and characterise the magnitude and timing
of seasonal ice dynamic variability. For a region close to the grounding line of Byrd Glacier, they
estimated seasonal variability of $\Delta U$ with a range of ~45 m a$^{-1}$, although this estimate is subject to
substantial uncertainty due to irregular, seasonally-biased sampling (see their Fig. 4). Our
ensemble using the $\Delta L_{B2L}$ representation (**Fig. 9e**) shows a phase that is consistent with Greene et
al. (2020); however, our modelled range in $\Delta U$ is always less than 10 m a$^{-1}$.

## 4    Sources of uncertainty in SSH and ice flow response

Our ice sheet modelling results suggest that seasonal variations in SSH beneath RIS are sufficient
to drive ice velocity variations of several metres per year over a large portion of the ice shelf when
using the $\Delta L_C$ and $\Delta L_{B2L}$ parameterisations to represent visco-elastic migration of the grounding
line. The modelled velocity variability generally decreases with increasing distance from the ice



front, although large variability is also associated with several major outlet glaciers flowing
through the Transantarctic Mountains. However, the correlation between model and GNSS
observations depends on the model initialisation, friction law, the grounding line parametrization,
and the source of the SSH forcing. In this section, we discuss the sensitivity of the model and the
uncertainty of each of these parameters.

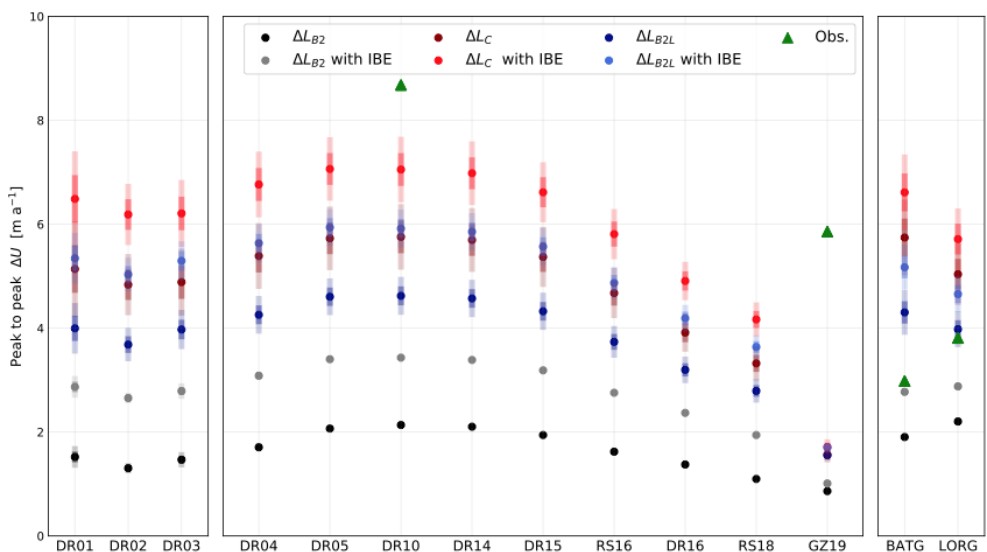


**Figure 10.** Peak to peak seasonal range of velocity anomaly ($\Delta U$) when forcing the ice flow model with

ocean model $\Delta SSH_{2002}$. The error bars (in shades) correspond to one and two standard deviations in each

ensemble. Observed peak to peak range is also plotted for GNSS stations with data records longer than one

year (i.e., DR10, GZ19, BATG, and LORG).


## 4.1    Model initialisation and friction law

The inverse model used to generate the initial steady state solution is under-constrained. Because
we infer two parameters with multiple constraints during the inversion, an initial state with a
minimal velocity misfit will not necessarily lead to a minimal ice thickness rate of change.





Different combinations of friction and viscosity parameters can lead to similar misfits. Using the
ensemble $\Omega_{15}$, consisting of the 15 optimal initial states (see Sec. 3.3 and Appendix A1) to estimate
the effect of the initialisation on the forward model helps quantify this effect. For the ensemble of
simulations using the $\Delta L_{B2}$ parameterisation, the impact of the initial state on the velocity seasonal
cycle is minimal over the ice shelf, with an average relative standard deviation under ~5% over
most of the ice shelf (**Fig. 8** (bottom row) and **Fig. 9 and 10**). The ensemble responses for the $\Delta L_C$
and $\Delta L_{B2L}$ parameterisations, while providing more realistic estimates of intra-annual velocity
changes, show more sensitivity to the initial state with year average relative standard deviation of
~15–20% ($\pm$0.1–0.15 m a$^{-1}$) at DR10 and ~25% ($\pm$0.4 m a$^{-1}$) at Byrd Glacier. We attribute the
relatively high variance of the ensemble in these regions to the sensitivity of the model to the initial
basal friction, while the relatively low variance of the ensemble over most of the ice shelf indicates
low sensitivity of the model to the initial viscosity parameter.

The friction law used in the model will also influence ice flow response, even for the same

value of $\Delta L$. Friction laws of different complexity have been proposed in the literature (Weertman,
1973; Budd et al., 1979; Schoof, 2005; Tsai et al., 2015), and have been shown to have different
impacts on grounding line dynamics (e.g., Brondex et al., 2019). We limited our study to the most
commonly used friction law (Weertman, 1973):
$\boldsymbol{\tau}_b = C \,|\boldsymbol{u}_b|^{\frac{1}{m}-1}\, \boldsymbol{u}_b,$                                    (2)
with $C$ being the friction coefficient, $\boldsymbol{u}_b$ the sliding velocity, and exponent $m \in [1 - \infty]$ where
increasing values of $m$ are characteristic of a more plastic bed. The results described in Sec. 3.2
were obtained with a linear version ($m = 1$) of Eq. (2); i.e., stress is proportional to velocity. We
also tested the value $m = 3$ (e.g., Brondex et al., 2019; Gudmundsson et al., 2019), which only
changes modelled velocity anomalies by a few percent. More complex friction laws (e.g., Schoof
et al., 2005; Tsai et al., 2015; Joughin et al., 2019) that include the impact of water pressure change
at the ice base as the grounding line migrates could increase the amplitude of our seasonal velocity
variations. However, such friction laws introduce additional poorly constrained parameters (Gillet-
Chaulet et al., 2016) and therefore, are not considered in this study.





### 4.2   Grounding line migration

The parameterisation of $\Delta L$ directly controls the amplitude of the grounding line migration which, in turn, controls the change in the friction coefficient we apply at the grounding line (see Sect 3.3 and Appendix B). $\Delta L_{B2}$ leads to small migration of the grounding line (typically a few metres), so that most of the impact of SSH variability on the ice flow comes from changes in $\Delta SSH$ gradients. While driving stress variations from these SSH gradients, and small grounding line migration ($\Delta L_{B2}$) due to $\Delta SSH$, can slow down or accelerate the ice flow by about 1 m a$^{-1}$ (**Figs. 6 and 7**), these modelled variations are only ~20% of the observed $\Delta U$ at DR10. Incorporating a larger grounding line migration in the model ($\Delta L_C$ and $\Delta L_{B2L}$) gives $\Delta U$ consistent with our GNSS observations. Such grounding line migration with respect to the hydrostatic case ($\Delta L_{B2}$) are, arguably, too strong, but are in line with observations by Brunt et al. (2011) and the values used by Rosier and Gudmundsson (2020) on Filchner-Ronne Ice Shelf. The surface and bed slope are key parameters of the grounding line migration parameterisation (Tsai and Gudmundsson, 2015; Appendix B). The bed slopes around the RIS perimeter, estimated by Brunt et al. (2011) by applying the hydrostatic assumption to observed migration of the inner margin of tidal ice flexure in repeat-track satellite altimetry, are likely to be biased low, based on the modelling of Tsai and Gudmundsson (2015). While the $\Delta L$ values given by $\Delta L_C$ and $\Delta L_{B2L}$ are in the upper range, they remain consistent with previous studies of tidal migration of the grounding line.

### 4.3   Estimating the Sea Surface Height Anomalies

The SSH anomalies ($\Delta SSH$) computed in the different ocean models (see Sec. 2.2 and SI) result from temporal variability in ocean currents driven by wind stress and lateral density gradients. However, these models do not account for the steric changes due to thermal and haline expansion and contraction, or the ocean's response to atmospheric pressure variations. Both the ROMS and NEMO modelling frameworks use the Boussinesq approximation based on the Navier-Stokes equations: the models conserve volume rather than mass and therefore do not properly account for steric changes. At the same time, variations of atmospheric pressure also lead to isostatic adjustments of the ocean (e.g., Goring and Pyne, 2003) while ice shelves have been shown to





respond similarly (Padman et al., 2003). This effect, known as the "inverse barometer effect"
(IBE), is also not considered in the simulations used in this study. Combining the effect of
Boussinesq SSH variations ($\Delta SSH_{boussinesq}$), the steric effect ($\Delta SSH_{steric}$), and the IBE
($\Delta SSH_{IBE}$), we obtain the total $\Delta SSH$ monthly deviation:
$\Delta SSH = \Delta SSH_{boussinesq} + \Delta SSH_{steric} + \Delta SSH_{IBE}$       (3)
Some efforts were made in the 1990s to evaluate the effect of steric sea level due to thermal
expansion, concluding that a globally uniform, time-dependent correction of sea level can correct
a non-Boussinesq solution (e.g., Greatbatch, 1994). Mellor and Ezer (1995) showed that the
seasonal variation of this term is about 1 cm in the Atlantic Ocean, which represents about 10% of
our modelled amplitude of SSH variation over the ice shelf. At the spatial scale of RIS, this
correction is roughly spatially uniform and, therefore, would not modify the driving stresses over
the ice shelf, but could affect the grounding line migration.
Seasonal changes in surface air pressure take place over the Antarctic continent, resulting in a
decrease of surface pressure (loss of atmospheric mass) from January to April and an increase of
surface pressure (gain of atmospheric mass) from September to December (Parish and Bromwich,
1997). Since most of the ocean models we presented use ERA-Interim reanalysis (Dee et al., 2011)
as an atmospheric forcing, we therefore use ERA-Interim surface pressure over RIS to estimate
the IBE effect contribution to $\Delta SSH$ and its potential effect on the ice flow. ERA-Interim is an
older product than the currently recommended ERA5-Land surface air pressure (Hersbach et al.,
2020) but both give similar surface pressures over RIS for the period we study, which limits the
uncertainty of the IBE effect.
We simulate the effect of IBE on $\Delta SSH$ following Eq. (1) and apply the full $\Delta SSH$ as a forcing to
the ice flow model. Due to the smaller isostatic adjustment of ice shelves to $\Delta SSH_{IBE}$ close to the
grounding line, we do not include its effect in the grounding line migration. The relative effect of
the IBE on the seasonal ice flow is maximal at DR10 and BATG due to their relative proximity to
the ocean. In contrast, GZ19 and the region of Byrd Glacier is less affected since the IBE does not
impact grounding line migration (**Fig. 10**). Overall, accounting for the IBE modifies the peak-to-



peak amplitude of ice flow variations by up to ~1.5 m a⁻¹ (**Fig. 10**) without significantly impacting
the seasonal pattern and phase of the ice flow velocity change. We note that, if the IBE was to have
a significant impact on the grounding line migration on average, it would most likely increase the
amplitude of the grounding line migration with a similar phase to the one we observe without IBE.
On a 38-year record of IBE (**Fig. S5**) the negative inverse barometer signal observed from
December to June would lead to downstream migration of the grounding line, and a slowdown of
the ice shelf. Conversely, the positive signal observed from July to November would lead to an
upstream migration of the grounding line, and an acceleration of the ice shelf.
**4.4    Ice rheology and time scales**
Our ice flow model uses the Shallow-Shelf Approximation (SSA), a viscous rheology, which is
well suited for studying long time-scale mechanisms involving ice creep (more than a few days).
At the same time, our parameterisations of the grounding line migration assume an elastic
rheology, which is more appropriate for short time-scale mechanisms such as tidal effects (less
than a few days). In reality, both rheologies are at play but either can sometimes be disregarded
with respect to the other, depending on the Maxwell time:
$t_M = \dfrac{\eta}{E}$,                                                                                        (4)
with $E$ the Young's modulus, and $\eta$ the characteristic viscosity of ice. Using a value $\eta \sim 40$ MPa
per year (obtained from the inferred viscosity parameter and strain rates averaged over the
ensemble $\Omega_{15}$ for RIS) and $E = 10^3 - 10^4$ MPa (Cuffey and Patterson, 2010) gives $t_M \sim 2$ days
to $\sim 2$ weeks. Given the seasonal time scale of the variability under consideration in this paper,
our viscous ice flow model adequately represents the real visco-elastic rheology of ice. The elastic
migration of the grounding line is, therefore, less representative of the actual visco-elastic rheology
for the time-scale changes we are observing (SSH anomalies remain relatively stable on periods
shorter than a month). However, the elastic parameterisation has previously been successfully
applied in a visco-elastic ice flow model studying ice flow response to fortnightly tidal forcing
(Rosier and Gudmundsson, 2020). Moreover, although the use of an elastic rheology to study a



viscous problem usually requires decreasing the effective Young's Modulus of ice (which could
decrease $\Delta L$), Tsai and Gudmundsson (2015) suggest that their parameterisation of the grounding
line migration may also apply to a purely viscous case. This could also explain why grounding-
line positions in Stokes models (that are not constrained to the hydrostatic approximation) are
generally more sensitive than in SSA models such as the one used in this study (e.g., Pattyn et al.,
2013).

## 5   Conclusions

We have used an ice sheet model to investigate our hypothesis that sea surface height (SSH)
variations can explain observed seasonal variability of ice velocity measured with four GNSS
records of roughly 1-2 year duration on Ross Ice Shelf (RIS). The model was forced with monthly
SSH fields obtained from ocean models that include thermodynamically active ice shelves.
Varying SSH fields can affect ice flow through two processes: changing the driving stress by
locally tilting the ice shelf; and by migration of the grounding line, which modifies the total friction
in the grounding zone. In ocean models that include ice shelves, the two sources of ice shelf
acceleration – surface SSH sloping downwards towards the ice front, and positive SSH anomalies
along the grounding zone **(Fig. 2b) –** are roughly in phase. We found that the ice sheet model is
able to reproduce the approximate phasing and magnitude of measured seasonal changes in ice
velocity, given appropriate parameterisation of visco-elastic processes close to the ice shelf
grounding line.
At seasonal time scales, changes in driving stress due to varying sea surface slope can only explain
10–20% of the observed range of ice velocity anomalies at the GNSS stations. However, if
grounding-line migration causes a sufficiently large change in friction in the grounding zone, our
ice sheet model generates seasonal ice flow signals with amplitudes and phases that are similar to
GNSS observations. The largest modelled annual changes in ice velocity under SSH forcing occur
along the ice front and close to the grounding lines of large glaciers flowing into the western RIS
through the Transantarctic Mountains. A large modelled seasonal cycle near the Byrd Glacier
grounding line is qualitatively consistent with, but much weaker than, satellite-based estimates by



Greene et al. (2020); however, these estimates and our models each have large uncertainties. In
the eastern RIS, intra-annual variability of ice velocity is generally weak; velocity changes
recorded by a GNSS (GZ19) station near the grounding line of Whillans Ice Stream are dominated
by a slowdown trend consistent with long-term trends of Siple Coast ice streams.
The modelled changes in bed stress due to grounding line migration as SSH changes depend on
parameterisation of visco-elastic processes. We considered two representations of these processes,
following Tsai and Gudmundsson (2015). Both provided similar responses at the GNSS station
locations (**Fig. 9**). When this parameterised migration is sufficiently large, the combination of
varying driving stress and grounding zone friction produces seasonal responses that are consistent
with the data records at the GNSS station locations (**Fig. 9**). Station DR10 in the central northern
RIS experienced the largest annual cycle, about 1% of the annual mean flow, while station GZ19,
located close to the grounding line of Whillans Ice Stream, does not include a substantial seasonal
cycle. Modelled annual ice flow changes at two stations in the northwestern RIS, BATG and
LORG, are smaller than at DR10 but still significant. There is some evidence in the data from these
sites to confirm the predicted annual cycles (**Fig. 9c,d**); however, these data records also include
substantial variability at ~6-month periodicity that is not apparent in the modelled signal. We
tentatively attribute this signal to the astronomically-forced, semi-annual variability in daily tidal
height range that results in time-averaged changes in grounded-ice flow through visco-elastic
processes (Rosier et al., 2020). However, in the absence of concurrent measurements of SSH
variability near the grounding line, we cannot rule out the presence of an SSH forcing signal with
~6-month periodicity that is not represented in the SSH forcing models. We note that ocean models
with annually repeating forcing, from which SSH forcing can be obtained, vary widely in their
estimates of seasonal variations (**Fig. S2**), while multi-year simulations with realistic forcing that
varies on interannual time scales produce large year-to-year changes in SSH (**Fig. S1**).
The largest modelled seasonal cycle in RIS ice flow occurs in the inlet close to the Byrd Glacier
grounding line (**Fig. 8, 9e**). There are no long-term GNSS records from this region to confirm the
modelled values; however, a previous study using satellite-derived variations in ice flow for Byrd
Glacier confirms that this region experiences large seasonal flow variability (Greene et al., 2020).





The high amplitude of the modelled velocity anomaly in this region is determined by the bed
geometry and the associated amplitude of the grounding line migration.
Our finding that seasonal signals in ice flow velocity are linked to SSH implies that improved
understanding of ocean-driven ice shelf velocity variations at intra-annual time scales can provide
valuable insights into the most efficient and accurate methods for modelling the likely future
dynamic response of ice shelves and grounded ice sheets as climate and sea level changes. Progress
is needed in four areas: (1) seasonally resolved measurements of open-ocean SSH; (2) ocean
modelling, including all components (mass, steric height change, and inverse barometer) that
contribute to SSH changes under ice shelves; (3) improved multi-year records of seasonally-
resolved ice velocity changes through either long-term continuous GNSS records or satellite-based
methods; and (4) representation of visco-elastic processes in the viscous models that, because of
computational limitations, are presently used for long time integrations of ice sheet processes.
Current satellite altimetry missions such as NASA's ICESat-2 can provide the SSH data close to
ice fronts for validating and improving ocean models of SSH including under ice shelves, while
concurrent GNSS measurements and reliable, data-constrained model estimates of sub-ice-shelf
SSH can be used to identify optimal configurations for viscous models and for tuning grounding
line parameterizations used in longer time integrations of ice shelf response to SSH changes.

**Appendix A: Inverse and direct ice flow model**
**A1. Ice flow model initialisation**
Following Klein et al. (2020), all our simulations were conducted at the scale of the RIS basin,
which encompasses the ice shelf and the grounded ice catchments that drain into RIS (Rignot et
al, 2011). We used a triangular finite element mesh with a spatial resolution that varies from 0.5
km at the grounding line to 20 km in regions of slow flow. The model spatial resolution on the ice
shelf is typically ~2 km. A Neumann condition, resulting from the hydrostatic water pressure
exerted by the ocean on the ice, was applied at the calving front (Gagliardini et al, 2013) and a





Dirichlet condition forced the normal velocities to zero on the inland boundary of the basins
adjacent to RIS.
Our model inversion optimises both the basal friction coefficient ($C$) and the effective viscosity of
the ice ($\eta_0$) by minimising multiple cost functions:
$$J_{total} = J_u + \lambda_{dh/dt} J_{dh/dt} + \lambda_C J_C + \lambda_{\eta_0} J_{\eta_0}$$    (A1)
where $J_u$ measures the difference between observed and modelled velocities, and $J_{dh/dt}$ measures
the misfit between modelled and observed thickness rates of change, computed as the difference
between flux divergence and mass balance (e.g., Brondex et al., 2019; Mosbeux et al., 2016). $J_C$
and $J_{\eta_o}$ are two regularisation functions added as constraints on the smoothness of the solution, by
penalising the first spatial derivatives of $C$ and $\eta_0$. Three of the four cost functions are weighted
by a regularisation parameter $\lambda$ to allow us to give more or less weight to a function.
We ran an ensemble of 100 inversions, varying the different regularisation parameters ($\lambda_{dh/dt}$,
$\lambda_C$, $\lambda_{\eta_0}$). The best members of the ensemble exhibit an ice flow pattern very close to observations,
with an RMS velocity misfit (RMS($\boldsymbol{u}$)) as low as ~10.1 m a$^{-1}$ and an RMS misfit on the ice
thickness rate of change (RMS($dh/dt$)) as low as ~0.7 m a$^{-1}$ over the grounded ice and the ice
shelf combined (**Fig. A1**). From this ensemble, we obtained a sub-ensemble of 15 members ($\Omega_{15}$)
with misfit values below 15 m a$^{-1}$ on velocities and 1 m a$^{-1}$ on ice thickness rate of change (**Fig.
A1**). Although this threshold on velocity is slightly higher than the data uncertainty reported by
Rignot et al. (2011, 2016), both thresholds are close to the RMS misfits in other studies based on
similar techniques (e.g., Gudmundsson et al., 2019; Brondex et al., 2019; Reese et al., 2018; Fürst
et al., 2015). This ensemble of initial states, $\Omega_{15}$, is then used for each of our simulations of
grounding line migration (i.e., $\Delta L_{B2}$, $\Delta L_C$ and $\Delta L_{B2L}$) for each model of SSH variability.

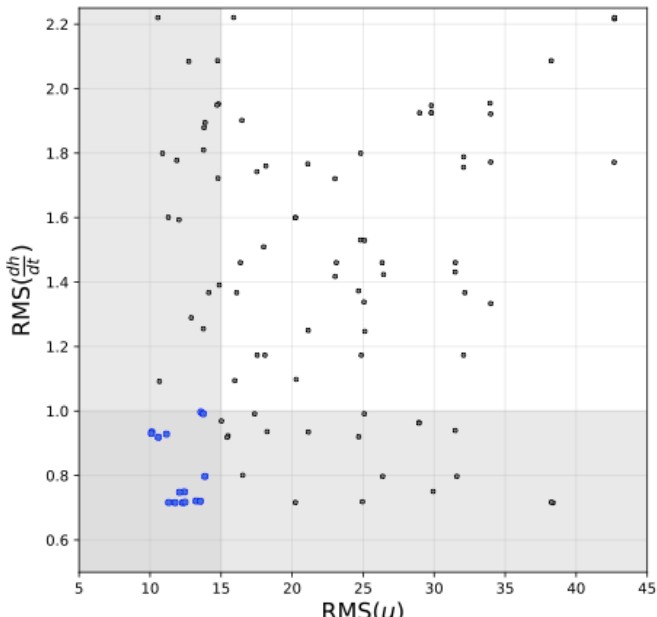

**Figure A1.** Ensemble of inversions (100 members, grey and blue points) in RMS(u) − RMS(dhdt) space.
The vertical and horizontal grey boxes represent the sub-spaces RMS(u) < 15 m a$^{-1}$ and RMS(dhdt) < 1 m
a$^{-1}$. The intersection of the two boxes represents the optimal sub-space ($\Omega_{15}$) which contains 15 members
(blue points).

## A2. On the use of a diagnostic ice flow model

Klein et al. (2020) reported that the initial state obtained after inversion is not perfectly stable
because of remaining uncertainties in other ice sheet parameters (see also, e.g., Seroussi et al.,
2011), which leads to locally large and unphysical ice thickness rates of change when running
transient simulations (e.g., Brondex et al., 2019; Gillet-Chaulet, 2012; Klein et al., 2020). This
problem is usually overcome by running a relaxation experiment, where the model is allowed to
evolve under a constant forcing until a more stable state is reached and before applying the desired
perturbation (e.g., Brondex et al., 2019; Gillet-Chaulet, 2012). However, this procedure sometimes
incurs a significant cost in terms of the differences between observations and the modelled ice
thickness and velocities. Although our initial states are similar to those in Klein et al. (2020), our





experiment differs by the nature of the perturbation we apply. The basal melting investigated by
Klein et al. (2020) directly affects the ice thickness, leading to a modification of the ice flow. The
SSH deviations used here do not directly modify the ice thickness but rather modify the driving
stress and grounding line position, which leads to a modification of the ice flow, eventually leading
to a dynamical change of ice thickness. These changes in ice thickness are fairly small and can be
neglected compared with changes in driving stress and grounding line position. Therefore, our
model does not actually vary in time; instead, we apply the monthly-averaged $\Delta SSH$ as a
perturbation to the Shallow-Shelf model and calculate the difference of the velocity field between
the perturbed model and the reference model. Monthly velocity change can therefore be
determined and compared with the GNSS velocity variations.
**Appendix B: Parametrization of the grounding line migration**
**B1. Theory and equations**
The grounding line migration under tidal variation is usually treated as a purely elastic and
hydrostatic problem (Tsai and Gudmundsson, 2015). In this context, the migration of the
grounding line can be formulated as follows:
$\Delta L^{\pm} = \frac{\Delta S^{\pm}}{\gamma^{\pm}},$           (B1)
where $\Delta S^{\pm}$ is the SSH perturbation in the grounding zone and
$\gamma^{+} = \beta + \frac{\rho_i}{\rho_w}(\alpha - \beta); \quad \gamma^{-} = \frac{\gamma^{+}}{1 - \rho_i/\rho_w},$           (B2)
with $\rho_i$ being the ice density, $\rho_w$ the water density, $\alpha$ the surface slope, and $\beta$ the bed slope.
The three parametrizations used in our study and presented in Sec 2.3.2 are further detailed here:
(i)      $\Delta L_{B2}$: we calculated $\Delta L_{B2}$ by applying $\gamma_{B2}$ values corresponding to Bedmap2 bed slopes

(e.g., $\beta_{B2} \sim [5 \times 10^{-3} - 5 \times 10^{-2}]$) and surface slopes (e.g., $\alpha_{B2} \sim \beta_{B2}/10$ on the ice

shelf and at the grounding line, and $\alpha_{B2} \sim \beta_{B2}/40$ when averaged over the entire basin)



in Eq. (B2), where $\gamma$ controls the length of the grounding line migration for a given $\Delta SSH$.

In the hydrostatic case, $\gamma_{B2}$, is calculated as a function of $\alpha$ and $\beta$.

(ii)    $\Delta L_C$: following Rosier and Gudmundsson (2020), we calculated $\Delta L_C$ by applying

constants for positive $\gamma^+{}_C = 5 \times 10^{-4}$ and negative $\gamma^-{}_C = \gamma^+{}_C /9$ bed slopes in Eq.

(B2).

(iii)    $\Delta L_{B2L}$: we calculated $\Delta L_{B2L}$ by applying a coefficient $\gamma_{B2L} = \gamma_{B2} / 20$, with $\gamma_{B2L}$ capped

to $\gamma_{B2L} = 1 \times 10^{-5}$ to limit extremely large grounding line migration in regions with very

small $\gamma_{B2L}$ values. This scaling factor was chosen so that the mean migration distance

around the RIS perimeter was similar to $\Delta L_C$

**B2. Subgrid-scale parametrization**
For $\Delta S^{\pm} = 10$ cm (roughly the maximal modelled $\Delta SSH$ for RIS), $\alpha = 5 \times 10^{-4}$ and $\beta =$
$5 \times 10^{-3}$ , Eqs. (B2) and (B3) lead to a $\Delta L^+ \sim 100$ m upstream and $\Delta L^- \sim 10$ m downstream
migration of the grounding line. These values are much smaller than the $\Delta x \sim 500$ m spacing of
our model grid nodes in the vicinity of the grounding line.
We overcome this problem by parameterizing the grounding line migration as a variation of the
friction coefficient at the grounding line (**Fig. B1**). Defining the basal shear force over the element
edges surrounding grounding line nodes as:
$F_i = \tau_i \, \Delta x$ ,                                  (B3)
with $\tau_i = C_i \, |u_i|^m$ , where $C_i$ is the reference friction coefficient and $u_i$ is the velocity on an
element edge, we can write the shear force over a fraction $\Delta x - \Delta L$ of the last grounded element
edge as:
$F_f = \tau_i \, (\Delta x - \Delta L)$.                               (B4)
Eq. (B4) can also be written as a function of a final shear stress integrated over the entire element:
$F_f = \tau_f \, \Delta x$                                      (B5)

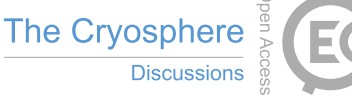

with $\tau_f = C_f \, |u_f|^m$ where $C_f$ is the friction coefficient at the grounded line node after migration
of the grounding line
Assuming $\left|\frac{u_f}{u_i}\right| \sim 1$, we can rewrite
$C_f = \frac{(\Delta x - \Delta L)}{\Delta x} \, C_i$                                                                   (B6)
with $C_f < C_i$ for $\Delta L > 0$ and $C_f > C_i$ for $\Delta L < 0$.

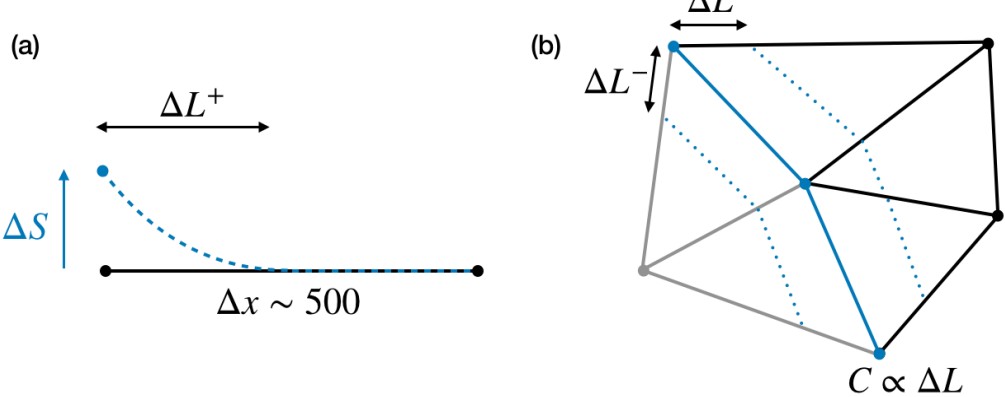


**Figure B1.** Schematic representation of the grounding line migration with sea surface height change ($\Delta S$).
(a) Fowline vue with $\Delta x$ the element edge size at the grounding line and $\Delta L^+$ the upstream migration of
the grounding line. (b) 2D-plan view of the virtual migration (dotted blue line) of the grounding line (blue
line) to an upstream ($\Delta L^+$) and downstream location ($\Delta L^-$); the evolution of the friction coefficient (C) is
proportional to $\Delta L^{\pm}$. Black and grey elements are initially grounded and floating.






*Code and data availability.* Elmer/Ice code is publicly available through GitHub (https://github.com/ElmerCSC/elmerfem; Gagliardini and others, 2013). All the simulations were performed with version 8.3 (Rev: b213b0c8) of Elmer/Ice. All Python 3 scripts used for simulations and post-treatment as well as model output are available upon request from authors. The data used are listed in the references.

*Author contributions.* CM, LP and HAF designed the study. CM conducted the simulations. CM and LP conducted the data analyses. EK and PDB provided the DRRIS data and provided insights in the interpretation of the data. All co-authors contributed to the writing of the paper.

*Competing interests.* The authors declare that they have no conflict of interest.

*Acknowledgement.* This research uses the data services provided by the UNAVCO Facility with support from the National Science Foundation (NSF) and National Aeronautics and Space Administration (NASA) under NSF Cooperative Agreement EAR-0735156 (GZ19) and EAR-1724794 (BATG). CM, LP and HF were supported by NASA grants 80NSSC20K0977, NNX17AG63G, and NNX17AI03G and by NSF grants 1443677 and 1443498. LP was also supported by NSF grant 1744789. PB was supported by NSF grants PLR-1246151 and OPP-1744856. GNSS data for GZ19 can be accessed at the UNAVCO data center (https://www.unavco.org/data/doi/ doi:10.7283/T53R0RPD). The IBE data were generated using Copernicus Climate Change Service Information [2020]. The modelling in this work used the Extreme Science and Engineering Discovery Environment (XSEDE), which is supported by NSF grant no. TG-DPP190003. The authors thank Richard Ray and colleagues for providing LORG GNSS data, and Scott Springer, Mike Dinniman, Kaitlin Naughten, Ole Ritcher, Pierre Mathiot and Nicolas Jourdain for providing SSH fields from their ocean models. The authors also thank Till Wagner, Pierre Mathiot and Nicolas Jourdain for their valuable comments and discussions on this manuscript.



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
