# Peer review of "Seasonal variability in Antarctic ice shelf velocities forced by sea surface height variations"

_The Cryosphere, 2022_

## Referee Comment (RC1)

**Review of *Seasonal variability in Antarctic ice shelf velocities forced by sea surface height variations* by Mosbeux et al. (2022)**

This study uses an ice sheet model to examine the impact of seasonal sea surface height (SSH) anomalies on ice flow of the Ross Ice Shelf. The SSH anomalies that are used to force the ice sheet model are derived from an ocean model that is verified against satellite altimetry measurements from 2018. Simulated changes in ice flow are compared with GNSS time series from various locations on the Ross Ice Shelf (RIS). The ice flow anomalies in response to SSH anomalies on seasonal timescales are on the order of metres per year, and the effect of grounding line migration is found to be the dominant process by which SSH anomalies induce changes in the ice flow response. The study argues that examining the impact of such short-term changes in SSH is important in being able to interpret the impact of future sea level rise on ice sheet dynamic response.

I found the manuscript an enjoyable read: it is well written, the figures are appropriate, and the methodology is sound and generally well-described. My main comment is on the implications and significance of the findings, which I detail below, followed by minor comments.

**Main comment**

Figures 7 and 8 show that the changes in ice flow in response to seasonal SSH changes are quite low ±3 m a$^{-1}$. From this, I wonder about how significant the impact of seasonal changes in SSH is? L25-26 of the abstract says: "... will provide further insights into longer term ice shelf and ice sheet response to future changes in sea level." It'd be great to see the significance of the findings discussed in the context of future sea level rise. Do you think including this SSH response at seasonal timescales is necessary to delineate between the ice flow response due to climate change compared with natural variability, for example? What kinds of uncertainties would we introduce by ignoring this process? On what timescales (e.g. for future ice sheet scenarios) is it important to consider these seasonal SSH processes? It'd be helpful to see a discussion of this in the manuscript, perhaps in Section 5.

To answer this, it could be helpful to consider whether/to what degree it's possible to scale the findings here to SSH variability on longer timescales? Without introducing new simulations, I'd be interested to see whether a relation could be found between the percentage changes in SSH to the percentage changes in driving stress to the percentage changes in ice flow (and similarly for the grounding line migrations). That is, is it possible to say for every XX% change in SSH, there is up to XX% ice flow change and XX km of grounding line migration? Would we expect these numbers to scale (and if so, linearly? Exponentially?) if we were, say, looking at interannual versus seasonal variability?

Although I'm not recommending this for the current study, it also might be interesting to test this by forcing the ice sheet model with sinusoidal variations in SSH, of different periodicities and amplitudes (e.g. that roughly approximate the seasonal cycle, ENSO, IPO, etc.), and see if a relationship does come out of it. This could be helpful in drawing implications for the impact of different modes of forcing compared with climate change-induced speed-up.

**Minor comments**

- L13-15: Sentence starting with "Ice shelf response…" could be made a bit more clear – it seemed to me to say the same thing twice – or even deleted.
- L14: provide → provides
- L24-end: Perhaps one more idea could be added here, which is how might we expect ice shelf velocity to respond with further percentage increases in SSH? Do we expect increases in SSH anomalies as the sea level rises?
- L42: "a few longer GNSS records and satellite-based estimates…show variability…" Would be good to include a citation here.
- L43-48: do you mean here that we could essentially scale-up what happens over the seasonal cycle to longer modes of climate variability? If so, it'd be good to explain this in a bit more detail.
- L62: "." → ","
- L68-69: "several GNSS records" → would be good to see citation
- L68: missing )
- L70: seasonal, annual, and intra-annual are used in different parts of the manuscript. Do these all refer to the same periodicity? Would be good to clarify.
- L84: suggest deleting "long" and just clarifying the time span
- Figure 1: would be great to see the domain of the ice sheet model. Would it be possible to include it on this figure?
- All figures: it would be great if the fontsize of all the text in the figures was at least the same size as that of the manuscript text. I found it a bit difficult to read some of the labels in the figures.
- L136: ", using" → ". We used" (break up long sentence)
- L137: "to determine the most realistic". What metric did you use to determine what is the most realistic?
- L148: modeling framework. I understand that the model setup and initialization is discussed at length in Klein et al. (2020). However, it would be very helpful to see a few more details here, either in the main body or in the supplementary. For example: the area that comprises the model domain, initialization datasets (ice surface elevation, topography), what grounding line is used, flow law and basal friction law (and example outputs of these fields).
- L149-150: I don't think I fully understood how the SSH variability is used to force the model. The SSH time series (over the full RIS spatial domain) is derived from the Tinto et al. (2019) model output, but which particular timestamps, and how many of them, are used to force the ice sheet model? Also, are these SSH anomalies applied as an increase/decrease in the ice shelf elevation (surface and draft)? A short description of that in the SI would be helpful.
- L153-154: It'd be good to include a discussion on the implications of using the SSA model here. As discussed in Rosier and Gudmundsson (2016), full Stokes is necessary to capture the flexural stresses associated with tidal motion, which are similar to the stresses induced in this model from seasonal SSH variability. Do you think this could have a significant impact on the results, including the extent of grounding line migration?
- L177: increase in driving → decrease in driving
- L178: an ice flow slowdown → ice deceleration (here and elsewhere – just a suggestion!)
- L179: an acceleration of the ice flow → ice acceleration (here and elsewhere – just a suggestion!)
- L207-208: some more info on these 100 inversions would be great. What parameter values did you vary? How did you choose how they should vary?
- L221: products → product

- L245: "shows a similar range of variability" to the DRRIS station?
- L261: is the underline on range a typo?
- L264: what filter did you use? Would be good to have a brief description here.
- Figure 5: Interested to hear your interpretation of the DRS data. How much does it impact the RIS signal? I imagine there's a much heavier weighting for the agreement between the RIS and OCS in determining which model is most realistic, but how does the DRS data come into the analysis or decision-making framework?
- L302-303: but in the same region where the absolute changes in SSH are larger? I.e. the yellow-er areas in figure 6c corresponding to the purple-er areas in figure 6b?
- L310: does "regionally-averaged" here refer to the whole ice shelf? Also, do certain regions have more weighting in this regional average, and if so which regions (presumably close to the ice-ocean front?)
- Figure 6: This is a small detail, but there's a line of bulls-eye like points across the RIS that is particularly evident in figure 6a and I've highlighted below from figure S4a (green). Is this something to do with the baseline surface elevation used in the model or something dynamical?

[Figure]

-
- Figure 7: Can we know from the simulations about how changes in which region of the ice shelf have the greatest impact on the ice surface speed? It'd be possible to test this with new simulations. E.g. if there was a series of experiments where the driving stress and grounding line migration processes were isolated to see how they each impact the ice surface speeds, and if so by what magnitude, for different regions of the shelf. These are probably outside the scope of the current study, but I can imagine it might be helpful to know that kind of information so that we can target specific regions for longer-term monitoring (e.g. deploy GNSS).

  Also, these speed anomalies are very small! How much should we care about these changes? Or how much would SSH have to vary before we cared? I can imagine small changes could be particularly important for some ice streams where migrations of the grounding line, even minor ones, could lead to more marked retreat. Is that the case for many of the ice streams that feed the RIS?

- L433: Do you have any thoughts on why the modeled range in ΔU is much less than the estimate from satellite altimetry?
- Figure 10: I found it a bit hard to tell the difference between some of the colors in this panel, particularly the blues. Would it be possible to use a greater contrast in the colors used? Another option would be to increase the thickness of the lines to minimize the white space and better distinguish between the different hues.
- L463-464: is there a relationship between the magnitude of the variance and particular features of the friction coefficients from the different inversions? How much do the different friction coefficients vary in the critical regions (e.g. near the grounding line of the ice stream?). Also, what values do you give the friction coefficient in regions that were floating during the inversion that become grounded due to grounding line advance?
- L487: decelerate or accelerate the flow by $\pm 10$ m a$^{-1}$? (add the $\pm$)
- L490-492: also, we do not expect that the ice is in hydrostatic equilibrium at the grounding line
- Section 5 Conclusions: overall I found the conclusions long, and could be made a bit more concise by removing some of the summary of the results that is a repetition from the results section (e.g. sentence over line 597-600), and instead a stronger focus could be made on the implications
- L618-632: As per main comment, it would be great to hear more here about how we can use the findings of this study to better understand the differences between the ice flow response to variability and climate change. Also, how much do you think it matters that we capture the impact of the seasonal cycle of SSH on ice shelves? How much uncertainty does its neglect introduce into model simulations, e.g. of future sea level rise? With increasing SSH, do we expect to see an increase in the magnitude of the SSH changes, and hence an increase in the significance of the processes examined in this study?
- L653: how did you choose appropriate ranges for the regularization parameters?
- L683: "are fairly small" → would be helpful to add an order of magnitude?
- Appendix B2. Subgrid-scale parameterization: I was a bit confused by this description, although I'm not very familiar with subgrid-scale parameterizations! It would be helpful for all the terms to be labeled. For example, what does the i subscript, and the Δx refer to?
- L734: fowline vue → flowline view?
- L954: the word Filchner-Ronne has some extra unintended characters in there! This occurs elsewhere in the references when special characters are being used, including letters with accents, and the years are missing from the references.

New reference

Rosier, S. H. R. and Gudmundsson, G. H. (2015): Tidal controls on the flow of ice streams, Geophysical Research Letters, 43 (9), 4433-4440, https://doi.org/10.1002/2016GL068220

---

## Author Response (AR1)

**Seasonal variability in Antarctic ice shelf velocities forced by sea surface height variations**

**-Response to reviewers-**

Mosbeux et al.

March 11th 2023

We would like to thank the two anonymous reviewers and the editor for their positive and constructive comments. We have addressed the different concerns of the Reviewers in this new version of the manuscript and believe that the manuscript is much improved for this editing. Our responses to Reviewer's comments will be colored in blue, directly following each comment.

**Reviewer 1**

*This study uses an ice sheet model to examine the impact of seasonal sea surface height (SSH) anomalies on ice flow of the Ross Ice Shelf. The SSH anomalies that are used to force the ice sheet model are derived from an ocean model that is verified against satellite altimetry measurements from 2018. Simulated changes in ice flow are compared with GNSS time series from various locations on the Ross Ice Shelf (RIS). The ice flow anomalies in response to SSH anomalies on seasonal timescales are on the order of metres per year, and the effect of grounding line migration is found to be the dominant process by which SSH anomalies induce changes in the ice flow response. The study argues that examining the impact of such short-term changes in SSH is important in being able to interpret the impact of future sea level rise on ice sheet dynamic response.*
*I found the manuscript an enjoyable read: it is well written, the figures are appropriate, and the methodology is sound and generally well-described. My main comment is on the implications and significance of the findings, which I detail below, followed by minor comments.*

We would like to thank the Reviewer for their thorough evaluation of our work and their appreciation. This feedback has allowed us to make substantial improvements to the manuscript. Below, we have aimed to address all general as well as minor comments

**Main comment**

*Figures 7 and 8 show that the changes in ice flow in response to seasonal SSH changes are quite low ±3 m a$^{-1}$. From this, I wonder about how significant the impact of seasonal changes in SSH is? L25-26 of the abstract says: "... will provide further insights into longer term ice shelf and ice sheet response to future changes in sea level." It'd be great to see the significance of the findings discussed in the context of future sea level rise. Do you think including this SSH response at seasonal timescales is necessary to delineate between the ice flow response due to climate change compared with natural variability, for example? What kinds of uncertainties would we introduce by ignoring this process? On what timescales (e.g. for future ice sheet scenarios) is it important to consider these seasonal SSH processes? It'd be helpful to see a discussion of this in the manuscript, perhaps in Section 5.*

It is true that the responses we observe in our models (and in the GNSS observations) are relatively small. We might have misled the reader/Reviewer when talking about the significance of our results. It is unlikely that SSH seasonal variations will lead to much more than ±1–10 m a$^{-1}$ of ice speed change. We think this is also why the process is detectable on Ross Ice Shelf, which is relatively

stable. Such change might be indiscernible on ice shelves with much more flow variability such as those in the Amundsen Sea. However, there are two ways in which we think our results are important, and we will make these clearer throughout the revised paper. First, the annual forcing provides a diagnostic tool for determining whether we are correctly representing the processes that will decide how future changes in SSH will affect ice flow. This is similar to how tides have been used to investigate ice and till physics near the grounding zone. Second, our study identifies features of the ice dynamics and bed that need to be better observed to improve how they are represented in models.

We don't feel that any significant extrapolation of our results is appropriate to long-term trends. However, if there is a change in seasonality of SSH, we should be able to estimate the associated change in seasonal ice response. This may be important in future if, for example, summer acceleration coincides and interacts nonlinearly with other seasonal forcings such as the near-ice-front basal melting investigated by Klein, Mosbeux et al. (2020; J. Glaciol.).
At the same time, note that the small seasonal SSH changes that we observe and model here are actually similar in amplitude to the annual rates of sea level rise that this ice shelf will experience in the future. Our results are directly relevant to what has been shown by Larour et al. (2019): "Slowdown in Antarctic mass loss from solid Earth and sea-level feedbacks". In their modelling, they show that sea level rate of change of about 10 cm a$^{-1}$ could affect the grounding line migration by about 40% with respect to models that do not include such processes. To make this clearer, we will add and comment on this reference in Section 5 of the new version of the manuscript.

*To answer this, it could be helpful to consider whether/to what degree it's possible to scale the findings here to SSH variability on longer timescales? Without introducing new simulations, I'd be interested to see whether a relation could be found between the percentage changes in SSH to the percentage changes in driving stress to the percentage changes in ice flow (and similarly for the grounding line migrations). That is, is it possible to say for every XX% change in SSH, there is up to XX% ice flow change and XX km of grounding line migration? Would we expect these numbers to scale (and if so, linearly? Exponentially?) if we were, say, looking at interannual versus seasonal variability?*

We agree that such a relationship would be really interesting. We could make a crude link between SSH-induced tilt and ice flow change for a very idealized case. However, the relation to the grounding line migration is not as easy to make since it also is strongly controlled by the basal friction in the grounding zone, which is spatially variable

*Although I'm not recommending this for the current study, it also might be interesting to test this by forcing the ice sheet model with sinusoidal variations in SSH, of different periodicities and amplitudes (e.g. that roughly approximate the seasonal cycle, ENSO, IPO, etc.), and see if a relationship does come out of it. This could be helpful in drawing implications for the impact of different modes of forcing compared with climate change-induced speed-up.*

This is indeed an interesting idea. Such simplification of the SSH signal could indeed help highlighting the relationships you mentioned. We think that it is also in line with recent studies by Robel et al. (2019) which shows that the inclusion of interannual to interdecadal variations of oceanic forcing significantly increases the uncertainty of future sea level rise projections. More recently, Felikson et al. (2022) showed that simulations accounting for seasonal variations of tidewater glacier terminus can affect positively or negatively (depending on the bed geometry and the amplitude of the oscillations) the mass balance of the glacier with respect to simulations neglecting these oscillations.

We think that, to apply your idea, we should even use a simplified ice shelf and ice sheet geometry so that our results could be more easily transferred to other regions and, maybe, other types of forcing.

**Minor comments**
- *L13-15: Sentence starting with "Ice shelf response..." could be made a bit more clear – it seemed to me to say the same thing twice – or even deleted.*

Yes, we agree that this sentence could be made clearer. We propose to rewrite it *"Ice shelves respond to changes in both the atmospheric and oceanic processes, each having large annual cycles; monitoring variability of ice velocity allows us to explore the processes by which environmental changes affect dynamics of both ice shelves and the buttressed grounded ice".*

- *L14: provide → provides*

Thanks for noticing the typo. We changed that.
- *L24-end: Perhaps one more idea could be added here, which is how might we expect ice shelf velocity to respond with further percentage increases in SSH? Do we expect increases in SSH anomalies as the sea level rises?*

This is a really interesting idea, thank you. We do think that SSH anomalies will change in future climates, since the seasonality of winds and thermal forcing will change. This is not necessarily directly coupled with sea level rise. While it is beyond the scope of our paper to explore expected changes in annual cycles, we will modify the Abstract and Conclusions to point others to this possibility. This adds to the idea of potential changes in seasonality of basal melting distributions discussed by Klein, Mosbeux et al. (2020; J. Glac.), especially as we expect seasonal upper-ocean warming to increase as sea ice volume decreases. Proposed new text:
*"We expect that climate-driven changes in the seasonal cycles of winds and upper-ocean summer warming will modify the seasonal response of ice shelves to SSH, and that nonlinear responses of the ice sheet will affect the longer trend in ice sheet response and its potential sea level rise contribution."*

- *L42: "a few longer GNSS records and satellite-based estimates...show variability..." Would be good to include a citation here.*

We added references to Klein, Mosbeux et al. (2020) for GNSS-based time series and to Greene et al. (2018, 2020) for satellite-based time series.

- *L43-48: do you mean here that we could essentially scale-up what happens over the seasonal cycle to longer modes of climate variability? If so, it'd be good to explain this in a bit more detail.*

Yes, we believe that there are many things to learn about specific processes from this observed intra-annual variability, such as the role of sea ice (*e.g.*, Greene et al., 2018), seasonal melt (*e.g.*, Klein, Mosbeux et al., 2020) and SSH variations (the hypothesis of this paper). However, our main idea here is that the availability and the repetition of seasonal observations give valuable insights on how ice shelves may respond to external forcings over longer timescales. We intend to clarify this by rewriting as follows: *"Given that the seasonal cycle dominates variability in atmospheric and oceanic forcing of ice shelves, understanding how this forcing affects ice shelf flow may provide important insights in the processes affecting the ice shelves and ice sheets, and how they might respond to the weaker but more persistent forcing that acts on longer time scales, from interannual variability (e.g.,*

*Dutrieux et al., 2014; Paolo et al., 2018) to multi-decadal trends (Jenkins et al., 2018)*." We hope this clarifies our thinking here, and if not, then we would also be happy to remove this statement from the manuscript.

- *L62: "." → ","*

Fixed.

- *L68-69: "several GNSS records" → would be good to see citation*

We added "(see Sec 2.1)" where the details about the GNSS stations and the appropriate references are cited.

- *L68: missing )*

Fixed.

- *L70: seasonal, annual, and intra-annual are used in different parts of the manuscript. Do these all refer to the same periodicity? Would be good to clarify.*

Thank you for pointing out this source of confusion. The "annual cycle" is intended as the repeatable pattern of forcing and response. "Seasonal variations" were intended to refer to ~3-month steps in this annual cycle. We used the expression, "intra-annual variations", to refer to any variability on a shorter-than-one-year time scale, regardless of whether it was related to the seasonal cycle of forcing. However, we can see how this leads to confusion, and so we will drop the latter terminology after providing a broader definition of "season variability".

- *L84: suggest deleting "long" and just clarifying the time span*

Done. We propose to say *"Several long (5-19 months) time series"*

- *Figure 1: would be great to see the domain of the ice sheet model. Would it be possible to include it on this figure?*

This is a good point; however, adding the entire domain to the figure would greatly reduce the size of the ice shelf on the page and would look a bit cluttered . As a compromise, we propose to add a figure of the entire domain (see example below) in the Supplementary Material. This will show the reader the increasing resolution of the mesh from the interior of the catchment towards  the grounding line; i.e., from ~25 km inland to ~500 m at the grounding line and ~2 km over the ice shelf.

[Figure]

**Figure S5: Finite Element mesh over the domain used for this study. The resolution varies from 25 km inland to 2 km over the ice shelf, and 500 m at the grounding line**

- *All figures: it would be great if the fontsize of all the text in the figures was at least the same size as that of the manuscript text. I found it a bit difficult to read some of the labels in the figures.*

We edited Figure 1, 2, 5, and 8 following your advice.

- *L136: ", using" → ". We used" (break up long sentence)*

Done.

- *L137: "to determine the most realistic". What metric did you use to determine what is the most realistic?*

We first made a visual comparison between the different model outputs, then performed a correlation test by studying the pixel-to-pixel correlation (R-correlation between the two dataset matrices). Figures 4 and 5 show the good correlation between the observations from the Armitage et al. (2018) altimetry-based dataset and the Tinto et al. (2019) model over the Open Continental Shelf. The details of the statistics are given in the Supplementary Material.

- *L148: modeling framework. I understand that the model setup and initialization is discussed at length in Klein et al. (2020). However, it would be very helpful to see a few more details here, either in the main body or in the supplementary. For example: the area that comprises the model domain, initialization datasets (ice surface elevation, topography), what grounding line is used, flow law and basal friction law (and example outputs of these fields).*

Thank you, some details were given in Section 2.3.1 as well as in the Appendix A1. Following your advice, we added some additional details in these sections. We summarise here a few of the key

choices of the modelling that we plan to include, so that you can check whether this is sufficient detail:

- Model Domain: we conducted at the scale of the RIS basin, which encompasses the ice shelf and the grounded ice catchments that drain into RIS (Rignot et al, 2011) as you suggested in a previous comment, we added a figure of the domain in our supplementary material.

- The SSA model uses an a vertically averaged effective ice viscosity with a nonlinear dependence on strain rate, and assuming isotropic material properties:

$$\eta = \eta_0 \varepsilon_e^{(1-n)/n},$$

  where, $\varepsilon$ is the second invariant of the strain-rate tensor, $\eta_0$ is a vertically integrated apparent viscosity parameter and $n = 3$ (Cuffey and Paterson, 2010).

- Bedrock elevation and ice thickness were taken from Bedmap2 (Fretwell and others, 2013), with a surface elevation correction applied to the floating ice to ensure flotation for an ice density of $\rho_i$ = 917 kg m$^{-3}$ and a water density of $\rho_w$ = 1028 kg m$^{-3}$.

- The friction law at the ice-bed interface is a Weertman's friction law (Weertman, 1973) and writes:

$$\boldsymbol{\tau}_b = C\,|\boldsymbol{u}_b|^{\frac{1}{m}-1}\,\boldsymbol{u}_b,$$

  with $C$ being the friction coefficient, $\boldsymbol{u}_b$ the sliding velocity, and exponent $m \in [1-\infty]$ where increasing values are characteristic of a more plastic bed. We made clear that our results use a linear friction law (m=1) and discussed this choice in the discussion 4.1.

- *L149-150: I don't think I fully understood how the SSH variability is used to force the model. The SSH time series (over the full RIS spatial domain) is derived from the Tinto et al. (2019) model output, but which particular timestamps, and how many of them, are used to force the ice sheet model? Also, are these SSH anomalies applied as an increase/decrease in the ice shelf elevation (surface and draft)? A short description of that in the SI would be helpful.*

We are sorry that we did not make this clearer the first time. The Reviewer is correct that the time series of spatially varying SSH is derived from the Tinto et al. (2019) model (chosen by comparison with altimetry-based SSH variability on the OCS; other models were also used and are presented in the SI). We used the Tinto et al. model data to construct monthly-averaged maps of SSH anomalies that were then used to force the ice sheet model. SSH anomalies are indeed applied as a rise or fall of the ice shelf surface and basal elevations. Some details about the process can be seen in the second paragraph of Section 2.3.2, which describes the model runs:

*"Using the sub-ensemble of initial state as a reference, we applied monthly averaged maps of SSH anomalies ($\Delta SSH(x,y)$) from five different ocean models (see SI) as a steady-state perturbation, raising or lowering the ice surface, and ice base and computing the flow change with respect to the reference (see Appendix A2)."*

- *L153-154: It'd be good to include a discussion on the implications of using the SSA model here. As discussed in Rosier and Gudmundsson (2016), full Stokes is necessary to capture the flexural stresses associated with tidal motion, which are similar to the stresses induced in this model from*

*seasonal SSH variability. Do you think this could have a significant impact on the results, including the extent of grounding line migration?*

Theoretically yes, the use of a SSA model, instead of a full Stokes model, will impact the solution. In an SSA model, the grounding line migration is usually hydrostatic. This is not necessarily the case in a full Stokes model where the grounding line position depends on the difference between the stress in the ice at the base and the water pressure. However, the length scale of the grounding line migration induced by the SSH variations is well under the spatial resolution of our model, so that without a proper subgrid parameterisation, the grounding line would not migrate at all.

In our case, we address the uncertainties in these representations by parameterising the grounding line migration following two hypotheses:

1. a small migration based on purely hydrostatic model (this is the most conservative case and represents the lower end member of our model ensemble)
2. a large migration that accounts for elastic effects at the grounding line (this directly follows Rosier et al. (2020) as well as the recommendation of Rosier and Gudmundsson (2016)).

Following the other Reviewer's comments on our large parameterised migration of the grounding line, we also added a discussion on other processes that can lead to a significant variation of the basal shear stress in the grounding zone.

- *L177: increase in driving → decrease in driving*

You are right, thank you for catching this inconsistency.

- *L178: an ice flow slowdown → ice deceleration (here and elsewhere – just a suggestion!)*

We went with "a deceleration of the ice flow" to match "an acceleration of the ice flow" in the same sentence. Our idea is to keep "flow" in the statement.
- *L179: an acceleration of the ice flow → ice acceleration (here and elsewhere – just a suggestion!)*

See last sentence.

- *L207-208: some more info on these 100 inversions would be great. What parameter values did you vary? How did you choose how they should vary?*

We basically vary the magnitude of 3 constraints: divergence, basal friction smoothness, integrated-viscosity smoothness. We will add the sentence *"The set of inversions explores the misfit between observed and model ice thickness variations, as well as basal-friction and integrated viscosity smoothness."* after lines L207-208. We will also add more details in the Appendix A1 to avoid cluttering the manuscript with technical details on the initialization (see the latter comment on L653).

- *L221: products → product*

Fixed.
- *L245: "shows a similar range of variability" to the DRRIS station?*

Yes, it shows a similar range of variability to the other DRRIS stations (although not seasonal). We have specified *"[…] shows a similar range of variability (+/- 1 m) to DR10 […]"*.

- *L261: is the underline on range a typo?*

Yes, it comes from our edits and discussion about the term we would use to qualify the amplitude/range of the tidal signal. The underline will be removed.

- *L264: what filter did you use? Would be good to have a brief description here.*

We used a sliding Gaussian filter with a 2-week standard deviation. This filtering results in the black line you can see in Fig. 3. We will make this clearer in the revised manuscript.

- *Figure 5: Interested to hear your interpretation of the DRS data. How much does it impact the RIS signal? I imagine there's a much heavier weighting for the agreement between the RIS and OCS in determining which model is most realistic, but how does the DRS data come into the analysis or decision-making framework?*

The Reviewer is correct; our choice of SSH forcing is mostly guided by the agreement between the models and the Armitage et al. (2018) altimetry-based observations over the OCS (Open Continental Shelf). Most models show similar performance over the DRS except NEMO, which performs better. This can be seen in the supplementary material in Fig. S3. However, NEMO performs relatively poorly over the OCS, showing much less variability than the CS2 observations. This can be explained by a sea-level correction applied to NEMO for global ocean simulations. This correction introduces a control of the mean sea level in order to prevent unrealistic drift of the sea surface height due to inaccuracy in the freshwater fluxes resulting from ice melt (from NEMO documentation and a personal communication with Pierre Mathiot who was in charge of the simulations).

- *L302-303: but in the same region where the absolute changes in SSH are larger? I.e. the yellow-er areas in figure 6c corresponding to the purple-er areas in figure 6b?*
-

Yes, there is indeed a direct link between the gradients in Fig. 6b and the driving stress change in Fig. 6c. We made this clearer by adding a similar statement to yours in line 308.

- *L310: does "regionally-averaged" here refer to the whole ice shelf? Also, do certain regions have more weighting in this regional average, and if so which regions (presumably close to the ice-ocean front?)*

Yes. We added *"(i.e., over the ice shelf)"* to make this clearer. No, we do not weight the regions of the ice shelf differently, the average is purely made by computing the mean of SSH values over the entire ice shelf, expressed as:

$\frac{1}{\Omega_{RIS}} \int \Delta SSH \, d\Omega_{RIS}$ , with $\Omega_{RIS}$ the surface of $\Omega_{RIS}$.

- *Figure 6: This is a small detail, but there's a line of bulls-eye like points across the RIS that is particularly evident in figure 6a and I've highlighted below from figure S4a (green). Is this something to do with the baseline surface elevation used in the model or something dynamical?*

This is directly linked to the surface DEM of the ice shelf, which retains texture from crevassing in the grounding zone as the ice flows towards the front. You can see a few of these structures, as well as larger rifts (see other areas presented in Fig. R1 below). LeDoux et al., (2017) noted that *"Many Transantarctic Mountains (TAM) outlet glaciers experience an abrupt change in flow direction as they enter the ice shelf. As a result, crevasse patterns due to shear around outlet corners can be short*

*lived, as ice turning past a corner enters into a region with different principal stresses".* This results in bumpy/crevassed areas similar to the suture zone that we observe downstream from Crary Ice Rise. We can also observe larger rifts in the middle of the ice shelf. We will add a note on this in the Figure caption: "*We can observe some large rifts as well as smaller scale surface structures such as crevasses*".

Such structures do not really have any effect in our modeling as their shape remains constant over our simulations, however displacements recorded on GNSS stations located close to the large rifts might be impacted by local deformations of the rifts (e.g. DR10 and DR14). This is mentioned in Klein, Mosbeux et al. (2020). For example, a flow-normal gradient was registered at DR10 and DR14. This might result from rift activity associated with shear stresses along the nearby rift tip near the suture zone, and with enhanced ice-quake activity in that region (Chen et al., 2019; Olinger et al., 2019).

[Figure]

*Figure R1. Along-flow gradient of ice shelf surface (see Figure S4 for more detail). We can observe some large rifts as well as smaller scale surface structures.*

- *Figure 7: Can we know from the simulations about how changes in which region of the ice shelf have the greatest impact on the ice surface speed? It'd be possible to test this with new simulations. E.g. if there was a series of experiments where the driving stress and grounding line migration processes were isolated to see how they each impact the ice surface speeds, and if so by what magnitude, for different regions of the shelf. These are probably outside the scope of the current study, but I can imagine it might be helpful to know that kind of information so that we can target specific regions for longer-term monitoring (e.g. deploy GNSS).*

Some of this, we feel, is covered by the present study. For example, the seasonal range of the SSH anomaly in the GZ is fairly constant around the RIS perimeter (Fig. 4) but Fig. 7 shows that acceleration through GZ migration effects is focused on glaciers coming through the Transantarctics. However, the Reviewer makes an excellent suggestion for future work. We see two options here:

1. For changes in GZ friction, we could follow the general approach used by Reese et al. (2018) to simulate key regions for basal melting: in our case, we could either simulate a change in SSH in different regions (Reese et al. use a change in thickness on patches of 20x20 km), or modify the friction for specific regions of the GZ, one at a time. However, the seasonal SSH field has a lot of spatial coherence, and the use of multiple ocean models for SSH to catch the variability might be a better way to look at realistic sensitivity.
2. use an inverse approach similar to Morlighem et al. (2021) and Baldacchino et al. (2022) where we would compute the gradient of velocity change over the ice shelf with respect to changes in SSH. This would give us a sensitivity map of the ice shelf to SSH. This method requires only one simulation but we would first need to adapt the inverse model implemented in Elmer/Ice.

As you mentioned, this is outside the direct scope of this paper but the second approach is in line with some work that we seek to develop in Elmer/Ice.

*Also, these speed anomalies are very small! How much should we care about these changes? Or how much would SSH have to vary before we cared? I can imagine small changes could be particularly important for some ice streams where migrations of the grounding line, even minor ones, could lead to more marked retreat. Is that the case for many of the ice streams that feed the RIS?*

We agree with you that both the observed and model anomalies that we analyze are relatively small in comparison with the average velocity of the ice shelf (on the order of 1%). Ross Ice Shelf (RIS) is probably one of the ice shelves where we have the highest chance to see these small signals because seasonal response to SSH changes seems to outweigh the response to seasonal ocean melting (Klein et al., 2020). Other ice shelves might experience larger changes but this would require a deeper analysis of the SSH in these regions as well as the potential grounding line migration (and subsequent basal conditions changes).

What we can say here is that:

(i) Even small variations provide insights into physical processes of the system that are important to the longer-term changes in the ice shelves. Unlike long-term trends, seasonal changes can be observed multiple times in available time series to provide some significant degrees of freedom for interpreting correlations.

(ii) some ice streams flowing into RIS, such as Byrd, seem to be "significantly" affected by the seasonal $\Delta SSH$.

(iii) even a minor increase in $\Delta SSH$ could increase the amplitude of the seasonal velocity change over a large portion of the ice shelf.

(ii) and (iii) imply that any attempt to identify subtle long-term changes in ice shelf dynamics through intermittent observations of ice velocity (e.g., through satellite-based feature tracking as in Greene et al., 2020) will be complicated by the possibility of an SSH-driven seasonal variability, that might also be modulated on interannual to decadal time scales by large-scale variability such as ENSO and SAM. We will emphasize these issues in the new version.

- *L433: Do you have any thoughts on why the modeled range in ΔU is much less than the estimate from satellite altimetry?*

We do have thoughts on this, and we plan to modify the text at line L433 to cite a few hypotheses to address this.

(1) the satellite data availability and quality is highly seasonal; see Figure 4 in Greene et al. (2020) and copied hereafter. There is a large scatter in velocity estimates from the multiple short-timescale differencing estimates available each summer, leading to very large uncertainty in defining the seasonal cycle. Currently, this is our working hypothesis, but it should be tested with GPS measurements on Byrd Glacier.

(2) We may be underestimating either the SSH signal in our model or the basal condition changes that are triggered by changes in $\Delta SSH$. (3) There might be other processes at play that we do not account for in our modelling; for example, the change in seasonal melt (explored in Klein et al., 2020) which, while expected to be small, could slightly increase the $\Delta U$ signal.

[Figure]

**Figure from Greeene et al. (2020). Velocity time series for an ITS_LIVE pixel. The clustering of these 14 208 measurements taken near the grounding line of Byrd Glacier typifies ITS_LIVE image pair data, with short-1t measurements providing direct but noisy observations of velocity variability throughout each summer, while much lower-noise winter estimates can only give insight into the total displacement that occurs during the dark, winter months. Light-gray horizontal bars connect the acquisition times of each image pair, and vertical bars show ±1σv uncertainty. Center dates tM are shown as dark-gray dots for visual clarity.**

Figure 10: I found it a bit hard to tell the difference between some of the colors in this panel, particularly the blues. Would it be possible to use a greater contrast in the colors used? Another option would be to increase the thickness of the lines to minimize the white space and better distinguish between the different hues.

We agree with you. We took both of these suggestions, and changed both the color and the thickness of the lines to help distinguish the different parameterisations. We also put the simulations with and without IBE on different vertical lines so that the error bars do not mask each other. The new version of the figure can be seen here:

[Figure]

**Revision of Figure 10.**

*L463-464: is there a relationship between the magnitude of the variance and particular features of the friction coefficients from the different inversions? How much do the different friction coefficients vary in the critical regions (e.g. near the grounding line of the ice stream?). Also, what values do you give the friction coefficient in regions that were floating during the inversion that become grounded due to grounding line advance?*

The value of the friction coefficient at the grounding line changes from one initial state to another, although they tend to all have a similar pattern. We expect some initial states to overestimate the friction at the grounding line (and underestimate the viscosity) and other initial states to underestimate the friction (and overestimate the viscosity).
We assume that regions that were floating (the nodes) and become grounded have the same friction as the region (the nodes) immediately upstream. This is a simple assumption that might overestimate the friction in the region since we expect the region to still be wet and slippery right after the ice enters in contact with the bedrock. However, our inversion scheme also accounts for the relatively low friction of the grounding ice in the vicinity of the grounding line.

- *L487: decelerate or accelerate the flow by ±10 m a$^{-1}$? (add the ±)*

Done.

- *L490-492: also, we do not expect that the ice is in hydrostatic equilibrium at the grounding line*

Yes, the Reviewer is right, we do not expect the ice shelf to be completely hydrostatic. As we stated in our manuscript, our large-scale grounding line migration is arguably too strong but we think that there might be other mechanisms at play in the grounding zone that could explain large basal shear stress change related to SSH variations. We detail this in our response to the other Reviewer.

- *Section 5 Conclusions: overall I found the conclusions long, and could be made a bit more concise by removing some of the summary of the results that is a repetition from the results section (e.g. sentence over line 597-600), and instead a stronger focus could be made on the implications*

Thank you for this comment. We shortened the conclusion and we also focused a bit more on the implications of the results, i.e. making our point clearer on the and how seasonal variations in SSH can inform us about the processes that will determine the future effect of sea level rise.

- *L618-632: As per main comment, it would be great to hear more here about how we can use the findings of this study to better understand the differences between the ice flow response to variability and climate change. Also, how much do you think it matters that we capture the impact of the seasonal cycle of SSH on ice shelves? How much uncertainty does its neglect introduce into model simulations, e.g. of future sea level rise? With increasing SSH, do we expect to see an increase in the magnitude of the SSH changes, and hence an increase in the significance of the processes examined in this study?*

As we stated in the previous comment, we tried to better answer these questions in our conclusion. We don't currently believe or propose that seasonality is a major component of uncertainty in long-term trends of grounded ice loss; however, (a) it does affect interpretation of subtle changes in ice velocity from intermittently-acquired data sets, and (b) there is a risk that nonlinearities could amplify the effect of seasonality in SSH-driven ice-shelf velocity, motivating further studies of those potential nonlinearities.

- *L653: how did you choose appropriate ranges for the regularization parameters?*

We tested a large range of values of regularization parameters:

$$\lambda_C = \{10^4, 5 \times 10^4, 10^5, 5 \times 10^5, 10^6\},$$

$$\lambda_{\eta_0} = \{10^4, 5 \times 10^4, 10^5, 5 \times 10^5, 10^6\},$$

$$\lambda_{dh/dt} = \{10^{-4}, 5 \times 10^{-4}, 10^{-3}, 5 \times 10^{-3}\},$$

which leads to $N_{simulations} = N_{\lambda_C} \times N_{\lambda_{\eta_0}} \times N_{\lambda_{dh/dt}} = 100$. These large ranges of values allow us to give very low to very high weight to each cost function, and well distribute the simulations in the parameter space (see Figure A1). In the end, we only kept inversions leading to low RMS misfit on the velocities and the thickness rate of changes, i.e., our sub-ensemble of 15 members.

We detailed the values of $\lambda_C, \lambda_{\eta_0}, \lambda_{dh/dt}$ in the new manuscript.

*L683: "are fairly small" → would be helpful to add an order of magnitude?*

Yes, we quantified the dynamic thinning/thickening triggered by the monthly grounding line migration and the monthly $\Delta SSH$. We have shown in Section 3.2 that gradients of $\Delta SSH$ range from $10^{-5}$ to $10^{-8}$ (for variations of a few centimeters in $\Delta SSH$). This is equivalent to changes in driving stress of 0.1-1% and even several percent in some areas (Fig. 6c).

Using our example of section 3.2, a tilting of 0.1 m at the front of the ice shelf leads to an overall change in driving stress of ~0.25%. In comparison, from a transient simulation, we have calculated

that the same changes in driving stress lead to dynamic thickness changes (thickening or thinning) of ~0.01%. We detail this in the Appendix of the revised manuscript.

- *Appendix B2. Subgrid-scale parameterization: I was a bit confused by this description, although I'm not very familiar with subgrid-scale parameterizations! It would be helpful for all the terms to be labeled. For example, what does the i subscript, and the Δx refer to?*

This subgrid-scale parameterisation is essentially geometrical and somewhat similar to the subgrid parameterisation you can find in the PISM model (e.g., Feldmann et al., 2014). Elmer/Ice does provide a more robust grounding line subgrid-scale parameterisation but only for the SSA model (2D).

The solution is to increase/decrease the basal stress at the grounding line proportionally to the increased/reduced sub-element migration, where the sub-element migration is parameterised as a function of the surface slope and the bed slope at the grounding line and the SSH variation (Eq. B1 and B2).

We will add some of the term definitions that were missing and probably led to some confusion and misunderstanding.

- *L734: fowline vue → flowline view?*

Fixed.

- *L954: the word Filchner-Ronne has some extra unintended characters in there! This occurs elsewhere in the references when special characters are being used, including letters with accents, and the years are missing from the references.*

Thank you for pointing this out. There was indeed a formatting problem with special characters. We will change this in the edited version of the manuscript. The publication year is written at the end of each reference. We checked that it was the case for all the references.

**Reviewer 2**

*This paper describes the effects of including annual sea-surface height changes when modelling the flow of ice shelves. By changing the height of the ice shelf, two changes to the flow are considered, firstly that raising the ice surface reduces the sea-ward driving stress and slows the flow, and secondly that raising the base of the ice causes the grounding line to retreat land-ward and reduces the basal drag, allowing the flow to accelerate. These contributions are quantified through the use of Elmer/Ice and the authors find that the change in grounding line position has a large impact on ice shelf velocity. It is a nice idea for a paper. But the models of grounding-line position are not correct for the timescales involved.*

*Looking at figure 2, the authors seem to be considering elastic flexure at the grounding line to be a major component of the ice shelf response, so that the direction of the surface perturbation close to the grounding line is opposite to that over the majority of the shelf (it is not clear what "relative uplift", l.307, actually means - relative to what? but I take it this is the effect being sketched). I cannot see any indication in figure 4 that this occurs - as the authors state, seasonal variations are much slower than the Maxwell timescale for ice, so the viscous relaxation should outweigh any elastic flexure, so the response of the shelf will primarily be that due to hydrostatic balance. In any case a rise in mean sea surface height should correspond to an inland migration of the grounding line (as stated in l.313); I should like to see figure 2 redrawn to remove the implication that the opposite occurs.*

*This brings me to my major concern - that the authors are using models for grounding line migration that were developed for a very different timescale, on which ice behaves primarily elastically. They attempt to justify this by reference to a paper that also uses this models for fortnightly behaviour - but that is an order of magnitude closer to the Maxwell timescale than the seasonal variations are. Elastic stresses within the ice will be negligible on seasonal timescales. I cannot really see why anything except hydrostatic balance would be appropriate here, and I cannot support publication of this paper while model (ii) is being given serious consideration.*

We thank the Reviewer for their nice and clear summary of our paper and for acknowledging the originality of our work. We also thank them for their important insights concerning the limitations of our elastic parameterization of the grounding line migration. We agree with them that the elastic model is inappropriate to use on seasonal timescales; our intent, however, was to attempt to bridge the gap between short-term (tidal-band) SSH effects where elastic is clearly dominant, and long-term SSH trends where viscous deformation dominates. We intend to leave it in as a high end-member of the possible effect of SSH variations on the ice flow, while being transparent about its limitations. We propose to make this clearer in the manuscript by including a few hypotheses that could explain the larger reduction in basal shear stress in the grounding zone needed to reproduce the observations of seasonal flow variations. The most important point is that the fully hydrostatic parameterisation leads to a small migration, whatever other assumptions we make.

Concerning the effect of SSH variations on the grounding line migration, the Reviewer is correct that Figure 2 depicts two effects at the same time: (1) an uplift/lowering of the ice front due to a relative positive/negative gradient of the SSH anomaly and (2) a downstream/upstream migration of the grounding line due to the negative/positive SSH anomaly at the grounding line. Figure 4 shows that these effects often combine. For example, over the period January-March, we see in the Tinto et al. (2019) model that $\Delta SSH > 0$ close to the ice front while it is <0 over most of the ice shelf and at the grounding line. We think that the sketch is representative of the seasonal variability of the

general mechanism we are observing for RIS (phasing of the two forcing mechanisms may vary on other ice shelves), with or without accounting for the elastic flexure of the ice shelf. However, it is true that we cannot assert that one effect necessarily leads to the other (e.g., a positive $\Delta SSH$ at the ice front does not necessarily imply a negative $\Delta SSH$ at the grounding line). We propose to make this clearer in the caption of Figure 2 by adding: *"Notice that the effects of ice shelf slope and grounding line position (positive/negative $\Delta SSH$ at the ice front and negative/positive $\Delta SSH$ at the grounding line) do not always act in the same direction, and sometimes their effects cancel. In that case, the net effect either accelerates nor decelerates the ice flow, depending on which effect is stronger"*.

Our large grounding-line migration assumes elastic flexure of the ice shelf under the SSH load, following Tsai and Gudmundsson (2015). Without this larger migration or, more precisely, without the equivalent large change in basal shear stress in the grounding zone, the modelled ice flow response is smaller than what we observe in the GNSS time series; however, it is still significant, as shown by the hydrostatic parameterisation B2 in Figs. 7 and 8). Our argument is that, even if the ice shelf tends to go back in a few days to a hydrostatic position after a perturbation of the sea surface height ($\Delta SSH$), the perturbation on the subglacial-network (wet and slippery bed/tills) could last much longer and weaken the basal shear stress for a longer period. Therefore, the elastic parameterisation of the grounding line can be viewed as a proxy to model a plausible range of seasonal variation of the subglacial hydrologic system and the associated basal shear stress variation.

We understand the reviewer's concern, and agree that our large scale migration model was missing some important context and justification. We still stress (both here and in the revised manuscript) that this parameterisation is a high end-member of the potential impact of $\Delta SSH$ on the ice flow. We agree that we should explore other mechanisms that could explain the large variations in basal shear stress needed to reproduce the observed amplitude of ice flow change. We describe two interlinked explanations below and plan to add them to the manuscript:

1. Our first explanation is linked to the relatively low value of the basal friction coefficients we inferred at the grounding line during the model initialization. Our model initialization relies on the optimization of the friction coefficient *C* in the Eq. (2) of the manuscript. We write this equation here:

    $$\boldsymbol{\tau}_b = C \, |\boldsymbol{u}_b|^{\frac{1}{m}-1} \, \boldsymbol{u}_b \,,$$

    with $C$ being the friction coefficient, $\boldsymbol{u}_b$ the sliding velocity, and exponent $m \in [1 - \infty]$ where increasing values of $m$ are characteristic of a more plastic bed. This law does not include a direct dependency on the effective pressure (like a Coulomb law would; e.g., Brondex et al., 2019; Urruty et al., 2022). However, as the friction parameter C is determined through inversion, it should include the dependency on the effective pressure and reduce the value of *C* at the grounding line to match observations (e.g., Urruty et al., 2022). The inferred friction represents an average annual value of the friction coefficient. The distribution of the seasonal variation around this annual average cannot be exactly determined without a proper knowledge of the subglacial hydrologic system, which is not realistically represented in ice-sheet modelling. However, one can assume that the variation could be larger than the variation we estimate through our hydrostatic parameterisation (i.e., a change in *C* directly proportional to the grounding line migration $\Delta L$). Seawater intrusion at the ice-bed interface and in sediments has been shown to have a high impact on the ice flow response (e.g., Robel et al., 2022). Subglacial models depending on subglacial water pressure decrease the effective pressure significantly near the grounding line, leading to an increased sensitivity for a given power in the sliding law

(e.g., Kazmierczak et al., 2022). Although the consequences of $\Delta L$ on this effective pressure is difficult to estimate, we believe that incorporating this mechanism in our modelling would lead to a larger impact of $\Delta SSH$ on the ice flow, even for the purely hydrostatic case.

Seawater intrusion could also be enhanced by a highly retrograde slope (e.g. Byrd Glacier in Fig. R2 of this Response to Reviewers). Retrograde bed slope will enhance both the migration of the grounding line and the intrusion of seawater in the subglacial hydrologic system.

[Figure]

**Figure R2. (a) Bed elevation with highlighted retrograde slope (sloping upward in the flow direction). (b) Grade (blue) and retrograde (red) bed slope in percentage (%). The mapping is based on the Bedmap2 dataset by Fretwell et al., (2013) and the direction of the ice flow computed during the initialisation phase.**

2. Our second explanation is directly linked to the potential effect of the grounding line migration on the subglacial water system. Such a mechanism would assume that *SSH* can vary over a short period (i.e. a few days) with a longer lasting effect on the subglacial hydrologic system in the grounding zone. We added a figure below that shows how the perturbation can evolve over September 2016 in MetROMS (Naughten et al., 2018; Fig. R3 of this Response to Reviewers). Each of these snapshots are separated by about 6 days, which is on the order of Maxwell time for ice, where both elastic and viscous effect matters. We can see that, while the average SSH of the month is positive in Naughten et al. (2018), the model rapidly switches from a low positive – low negative anomaly to a much stronger positive anomaly. This rapid change can lead to large elastic migration of the grounding line, in agreement with other studies (e.g., Tsai and Gudmundsson, (2015; Rosier et al.2015, 2016, 2020; Warburton et al., 2020). We recognise that in our case, the model does not switch back as fast to a negative anomaly as it would for a tidal loading. However, even if the grounding line quickly relaxes to its hydrostatic position due to the viscous relaxation, the perturbation of the hydrological system and the consequent weakening of the basal shear stress could last longer and extend over a long distance upstream the grounding line. Full treatment of the subglacial water system is out of the scope of this study, but could help validate our theories in the future. To some extent, this weakening is parameterised in our model.

Our large scale parametrisations of the grounding line migration ($\Delta L_C$ and $\Delta L_{B2L}$) tend to accommodate these two hypotheses, with intrinsic limitations to the exercise. We made this clearer in the manuscript by adding some of the details we gave here and emphasized that the basal drag change at the grounding zone is both a parameterisation of the grounding line migration itself and of the potential effect of the migration on the subglacial water system.

[Figure]

*Figure R3. Snapshots of daily $\Delta SSH$ (with respect to annual mean) in September 2016 in MetROMS (Naughten et al. 2018). The snapshots are for the 1st, 7th, 13th, 19th, 25th of the months.*

We are grateful that the Reviewer raised this concern, as we think it will considerably strengthen our manuscript. We believe that our explanations and planned modifications to the manuscript will address most of the Reviewer's concerns.

*A further concern regarding the model for hydrostatic grounding line position being used (since this is rather key to the remaining results) - the result of the Tsai and Gudmundsson paper, that downstream migration is 9 times less than upstream migration, assumes that the ice surface gradient is constant across the grounding line, while the gradient in ice thickness changes abruptly (by this factor of 9). If one makes the opposite assumption, that the ice thins uniformly through the grounding zone (e.g. Sayag and Worster 2011, Warburton et al. 2020), then the hydrostatic migration distance is completely symmetric. With access to all the data needed to test these assumptions, I would be more reassured if the authors calculated the hydrostatic migration distance "from scratch", rather than wholesale apply this massively idealised formula.*

Thank you for raising this source of misunderstanding. The main point of the hydrostatic migration is that it always leads to small grounding line migration, whatever the assumption you make when parametrizing it. The parameterisation we use in the manuscript is an asymmetric hydrostatic grounding line migration (Equations B1 and B2), and does indeed come from the fact that we use the same theory as Tsai and Gundmundsson (2015) (their equations (1)-(3)). For completeness, and to clarify our approach for the Reviewer, we rewrite here the trigonometrical construction on which this parametrization relies, as well as our understanding of the Warburton et al. (2020) assumption. The steps are as follows:

(i) At the grounding line, the ice is lifted due to floatation and the upward buoyancy force in the water column is compensated by the downward gravitational force in the ice column:

$$F_i = F_w \Leftrightarrow \rho_i \, g \, H = \rho_w g \, h_w = \rho_w(z_{SL} - z_b),$$

where $z_{SL}$ is the sea level and $z_b$ is the bed elevation.

(ii) Adapting Tsai and Gudmundson (2015), upstream the grounding line (GL), we can approximate the bed elevation at the point of migration of the GL ( zb,L ) by:

$$z_{b,\Delta L} = z_{b,GL} + \beta\, \Delta L,$$

with the bed slope (equal to the ice base slope if located upstream the GL) and L the GL migration we try to estimate. Similarly, the ice thickness upstream the grounding line can be estimated as:

$$H_{\Delta L} = H_{GL} + (\alpha - \beta)\, \Delta L \,,$$

(iii) From there, we can rewrite:

$$\frac{\rho_i}{\rho_w}(H_{GL} + (\alpha - \beta)\, \Delta L) += \Delta SSH - Z_{b,GL} - \beta\Delta L$$

and estimate

$$\Delta L^+ = \frac{\Delta SSH}{\frac{\rho_i}{\rho_w}(\alpha - \beta) + \beta}$$

(iv) For the downstream migration, as the Reviewer stated, our assumption leads to a reduction of the ice base slope by a factor $1/(1 - \frac{\rho_i}{\rho_w}) \sim 9$ and therefore a potential grounding line migration:

$$\Delta L^- = \Delta L^+ \times (1 - \frac{\rho_i}{\rho_w}).$$

(v) In Warburton et al. (2020), a constant thickness (named D in their manuscript, second line of sec. 2.1) is assumed. We can read "*For simplicity, we consider a constant ice thickness D across the grounding zone*" in their manuscript. If we are correct, this means that they assume no thinning through the grounding zone. In this case, and given a constant bed slope, the Reviewer is right that there would be no asymmetry in the grounding line migration but this would also mean that upstream the grounding line, $\alpha = \beta$.. The upstream migration would therefore be:

$$\Delta L^+ = \frac{\Delta SSH}{\beta},$$

and a similar downstream migration $\Delta L^- = \frac{\Delta SSH}{\beta}$ only if we assume that the bed is constant both upstream and downstream the grounding line.

(vi) In the end, it is true that both assumptions are relatively idealized, and in fact both yield a similar result. In the case where we assume $\alpha = \beta$, the Tsai and Gundmundson parameterisation leads to:

$$\Delta L^+ = \frac{\Delta SSH}{\frac{\rho_i}{\rho_w}(\alpha - \beta) + \beta} = \frac{\Delta SSH}{\beta},$$

similarly to Warburton et al. (2020). The asymmetry on L- in Warburton et al. (2020) depends on the bed slope downstream the grounding line. Assuming a constant bed slope, there would be no asymmetry and the bed would be

$$\Delta L^- = \frac{\Delta SSH}{\beta}.$$

(vii) In our parametrization, the Reviewer is correct that we assume the surface and bed slope to be constant in the grounding zone. Tsai and Gudmunsson mention the fact that *"average surface and bed slopes are potentially different immediately upstream versus immediately downstream of the grounding line, the differences are unlikely ever to be ~10 times different, so this result suggests that grounding-line migration over the positive part of the tidal cycle (high tide) dominates the migration over the negative part of the tidal cycle (low tide)"*. We think that this is especially true for small migrations such as the ones of our hydrostatic model $\Delta L_{B2}$ , i.e. about a few tens of meters except in some areas of the Siple Coast and some Trans-Antarctic glaciers (Figure R4a of this R2R), a length scale under our model resolution and on which we do not expect large surface and bed slope variations.

We agree that this is a limitation of our model and we will add this to our discussion. In the end, the $\Delta SSH$ -induced migration of the grounding line and the associated basal shear stress change of the hydrostatic case are small, with the Siple Coast also exhibiting a relatively low basal friction, limiting the effect of the migration and basal shear stress change on the ice flow (Figure R4b of this R2R). We emphasize to the reader about the small effect of $\Delta L_{B2}$ in our manuscript: *"We regard the $\Delta L_{B2}$ parameterisation, which yields small grounding line migration, as an approximation of ice shelf response to SSH gradients alone."* We will also add Figure 4 to the supplementary material to show the effect of our parameterisation. We hope that our explanations as well as this Supplementary figure will alleviate the Reviewer's concerns.

[Figure]

**Figure R4. (a) Migration $\Delta L_{B2}$ of the grounding line for $\Delta SSH = 5 \times 10^{-2}$ m. (b) Basal shear stress $\tau_b$ at the grounding line averaged over the ensemble of simulation $\Omega_{15}$.**

**Smaller comments**

*Given the inherent non-linearities of ice shelf dynamics (and indeed grounding line motion), to what extent is it valid to compare the average of a function (mean velocities over a month) to the function of an average (ice shelf model forced by mean SSH)? The authors could consider applying the same process with much more of the daily signal kept in the forcing, and then average the output over a month, to see if this differs from the model output from the monthly average.*

We agree with the Reviewer that complex processes related to ice shelf dynamics, changes in bed lubrication and grounding line migration, as well as tidal effects, are at play in the mechanism we observe and model. However, these non-linearities are hard to account for, and doing so is well beyond the goal of this paper. As observed by Warburton et al. (2020) in the context of tidal grounding-line migration: *"The origin of this nonlinear response of the surface velocity remains enigmatic".* This is also why we relied on a simpler parameterisation of the grounding line migration (and basal shear stress change).

*For clarification, in figure 9, is this one set of simulations per month, with a continuous line drawn between these points, or is the model forced with daily values of a monthly running average?*

Yes, the Reviewer is right. It is one set of simulations per month with a continuous line drawn between these points. For clarification, we will add dots for each month in the revised figure.

**Additional references**

- Feldmann, J., Albrecht, T., Khroulev, C., Pattyn, F., and Levermann, A.: Resolution-dependent performance of grounding line motion in a shallow model compared with a full-Stokes model according to the MISMIP3d intercomparison, J. Glaciol., 60, 353–360, https://doi.org/10.3189/2014JoG13J093, 2014.
- Felikson, D., Nowicki, S., Nias, I., Morlighem, M., & Seroussi, H.: Seasonal tidewater glacier terminus oscillations bias multi-decadal projections of ice mass change. Journal of Geophysical Research: Earth Surface, 127, e2021JF006249, https://doi.org/10.1029/2021JF006249, 2022.
- Kazmierczak, E., Sun, S., Coulon, V., and Pattyn, F.: Subglacial hydrology modulates basal sliding response of the Antarctic ice sheet to climate forcing, Cryosphere Discuss., 1–24, https://doi.org/10.5194/tc-2022-53, 2022.
- Larour, E., Seroussi, H., Adhikari, S., Ivins, E., Caron, L., Morlighem, M., and Schlegel, N.: Slowdown in Antarctic mass loss from solid Earth and sea-level feedbacks, Science, 364, https://doi.org/10.1126/science.aav7908, 2019.
- LeDoux, C. M., Hulbe, C. L., Forbes, M. P., Scambos, T. A., and Alley, K.: Structural provinces of the Ross Ice Shelf, Antarctica, Ann. Glaciol., 58, 88–98, https://doi.org/10.1017/aog.2017.24, 2017.
- Robel, A. A., Seroussi, H., Roe, G. H.: Marine ice sheet instability amplifies and skews uncertainty in projections of future sea-level rise, 116, 30, 14887-14892, Proceedings of the National Academy of Sciences, https://www.pnas.org/doi/full/10.1073/pnas.1904822116, 2019
- Robel, A. A., Wilson, E., and Seroussi, H.: Layered seawater intrusion and melt under grounded ice, The Cryosphere, 16, 451–469, https://doi.org/10.5194/tc-16-451-2022, 2022.
- Urruty, B., Hill, E. A., Reese, R., Garbe, J., Gagliardini, O., Durand, G., Gillet-Chaulet, F., Gudmundsson, G. H., Winkelmann, R., Chekki, M., Chandler, D., and Langebroek, P. M.: The stability of present-day Antarctic grounding lines – Part A: No indication of marine ice sheet

instability in the current geometry, Cryosphere Discuss., 1–34, https://doi.org/10.5194/tc-2022-104, 2022.

- Warburton, K. L. P., Hewitt, D. R., and Neufeld, J. A.: Tidal Grounding-Line Migration Modulated by Subglacial Hydrology, Geophys. Res. Lett., 47, e2020GL089088, https://doi.org/10.1029/2020GL089088, 2020.